# The GRISLI ice sheet model (version 2.0): calibration and validation for multi-millennial changes of the Antarctic ice sheet

Aurélien Quiquet[1], Christophe Dumas[1], Catherine Ritz[2], Vincent Peyaud[2], and Didier M. Roche[1,3]

[1]Laboratoire des Sciences du Climat et de l'Environnement (LSCE), UMR8212, CEA/CNRS-INSU/UVSQ, Gif-sur-Yvette Cedex, France
[2]Université Grenoble Alpes, CNRS, IRD, IGE, F-38000 Grenoble, France
[3]Earth and Climate Cluster, Faculty of Science, Vrije Universiteit Amsterdam, Amsterdam, the Netherlands

*Correspondence to:* A. Quiquet (aurelien.quiquet@lsce.ipsl.fr)

**Abstract.**

In this paper we present the GRISLI (Grenoble Ice Sheet and Land Ice) model in its newest revision (version 2.0). Whilst GRISLI is applicable to any given ice sheet, we focus here on the Antarctic ice sheet because it highlights the importance of grounding line dynamics. Important improvements have been implemented in the model since its original version (Ritz et al., 2001). Notably, GRISLI now includes a basal hydrology model and an explicit flux computation at the grounding line based on the analytical formulations of Schoof (2007) or Tsai et al. (2015). We perform a full calibration of the model based on an ensemble of 300 simulations sampling mechanical parameter space using a Latin Hypercube method. Performance of individual members is assessed relative to the deviation from present-day observed Antarctic ice thickness. To assess the ability of the model to simulate grounding line migration, we also present glacial-interglacial ice sheet changes throughout the last 400 kyr using the best ensemble members taking advantage of the capacity of the model to perform multi-millenial long-term integrations. To achieve this goal, we construct a simple climatic perturbation of present-day climate forcing fields based on two climate proxies, both atmospheric and oceanic. The model is able to reproduce expected grounding line advances during glacial periods and subsequent retreats during terminations with reasonable glacial-interglacial ice volume changes.

## 1 Introduction

Continental ice sheets are a major climatic component for Earth system dynamics. They operate on a variety of timescales, from diurnal to multi-millenial, through multiple feedbacks such as temperature - surface albedo, gravity waves and oceanic circulation changes related to freshwater flux release. Over the last decades, observations of the Greenland and Antarctic ice sheets (e.g. altimetry, gravimetry, echo sounding) have shown important changes such as an increase in surface and sub-shelf melt, glacier speed-up, dynamical thinning and drastic calving events (e.g. Mouginot et al., 2015; Paolo et al., 2015; Hall et al., 2013; Pritchard et al., 2009). While the two current ice sheets have been mostly stable for at least the last 1000 years, they are expected to contribute to future sea level rise, albeit with a largely uncertain magnitude. Conversely, in the past, there is evidence of sea level rise as fast as four metres per century (e.g. Fairbanks, 1989; Hanebuth et al., 2000; Deschamps et al., 2012). The study of well-recorded past events can help us to constrain the fate of the ice sheets in a warming Earth and to

disentangle the role of the different processes leading to rapid ice sheet destabilisation. The surface mass balance-height feedback has often been proposed as a major driver for rapid northern Hemisphere ice sheet disintegration (e.g. Abe-Ouchi et al., 2013; Gregoire et al., 2012). Another source of instability, for marine ice sheets such as the palaeo Kara-Barents or present-day Antarctic ice sheets, is related to the fact that large parts of the bedrock presents a retrograde slope from the grounding line.

Isostatic adjustment can, for instance, deepen the bedrock well below sea level. Additionally, glacial troughs may form a deepening bedrock from the grounding line towards the ice-sheet centre. Such bed properties lead to the marine ice sheet instability (MISI, Weertman, 1974; Schoof, 2007) responsible for an irreversible retreat of the grounding line in response to an initial perturbation such as local sea level change and/or increase in basal melting rate below ice shelves. The latter two processes are expected to play a crucial role for the stability of the Antarctic ice sheet in the future (e.g. Favier et al., 2014). Additional

instabilities may also occur on neutral/prograde bed slopes in relation to structural instabilities of tall ice cliffs (marine ice cliff instability, MICI, Pollard et al., 2015).

Continental ice sheets are difficult to model because they include processes operating on a variety of temporal, from diurnal to multi-millenial, and spatial scales, from a few metres to thousand of kilometres, but also because we lack crucial observations

(e.g. basal conditions and internal thermo-mechanics). Most numerical models consider ice sheets as an incompressible fluid, where motion can be described with the Navier-Stokes equations. Even if some processes have generally to be parameterised (e.g. ice anisotropy), the complete set of equations can be solved explicitly and does not require the use of any approximation. The most comprehensive continental ice sheet models, namely the Full-Stokes models, solve explicitly all the terms in the stress tensor (Gillet-Chaulet et al., 2012; Larour et al., 2012). Recent applications are promising but, due to computational

cost, continental scale applications are currently limited to a few centuries. As such, they are not yet the most suitable tool for palaeo-reconstructions or multi-millenial future projections.

In order to decrease the degree of complexity, simpler models were historically developed that make use of the small aspect ratio of ice sheets (vertical to horizontal scale ratio) to derive approximations for the Navier-Stokes equations (e.g. Hindmarsh,

2004). Such models are computationally much cheaper than Full-Stokes models, allowing for multi-millenial integrations. They are well-suited to study slow feedbacks such as glacio-isostasy or the impact of temperature and surface mass balance perturbations. The Grenoble ice sheet and land ice (GRISLI) model is one of these simpler models (Ritz et al., 2001). GRISLI consists of the combination of the inland ice model of Ritz (1992) & Ritz et al. (1997) and the ice shelf model of Rommelaere and Ritz (1996), extended to the case of ice streams treated as *dragging ice shelves*. GRISLI was in the late nineties the first

large scale ice sheet model with an hybrid shallow ice / shallow shelf system of equations. Whilst since Ritz et al. (2001) the fundamental equations for ice dynamics have not drastically changed, the model has nonetheless benefited from numerous contributions. To date, 30 papers published or in peer-review discuss GRISLI model simulation results. The range of applications have been very wide ranging from ice sheet reconstructions for deep-time palaeo-climate (e.g. Benn et al., 2015; Donnadieu et al., 2011; Ladant et al., 2014) and Quaternary (e.g. Peyaud et al., 2007; Alvarez-Solas et al., 2013; Quiquet et al., 2013;

Colleoni et al., 2016) to future sea level rise projections (e.g. Ritz et al., 2015; Peano et al., 2017). GRISLI has participated in

several inter-comparison exercises (Calov et al., 2010; Edwards et al., 2014; Koenig et al., 2015; Goelzer et al., 2017) and has been coupled to climate models of various complexities (e.g. Philippon et al., 2006; Roche et al., 2014; Le clec'h et al., 2018).

The aim of our current study is to provide a technical description of the GRISLI model in its current version (GRISLI version 2.0, hereafter GRISLI), including several additional features from Ritz et al. (2001). In particular, we have now included an explicit flux computation at the grounding line following the analytical formulation from Schoof (2007) and Tsai et al. (2015) in order to have a better representation of the grounding line migration. We also provide details on some components (sub-glacial hydrology and tracking particle scheme embedded in GRISLI) which are currently not documented in international scientific journals. In addition, we present a simple calibration of the mechanical parameters suitable for multi-millenia integrations and we show an example of the model response to glacial-interglacial forcing.

In Sec. 2 we describe the fundamental equations of the GRISLI ice sheet model with a particular emphasis on the model developments departing from Ritz et al. (2001). In sec. 3, we present a simple calibration methodology which aims at reproducing the observed present-day Antarctic ice sheet geometry. In Sec. 4, we discuss the ability of the model to simulate the Antarctic ice sheet changes over the last four glacial-interglacial cycles.

## 2 The GRISLI ice sheet model

GRISLI constitutive equations were presented in Ritz et al. (2001) and we aim here at giving a broad comprehensive description of the current model version, with its latest functionalities. Major model parameters are listed in Tab. 1.

### 2.1 Ice thermo-mechanics

#### 2.1.1 Ice deformation and mass conservation

Ice deformation and mass conservation in GRISLI version 2.0 is mostly treated as in Ritz et al. (2001) with the notable exception of the use of a polynomial flow law with the introduction of a linear, Newtonian, viscosity.

GRISLI considers the ice sheet as solely formed of pure ice with a constant and homogeneous density ($\rho$). In this approximation, the ice being considered as an incompressible fluid, the mass conservation equation can be written as:

$$\nabla.\mathbf{u} = \frac{\partial u_x}{\partial x} + \frac{\partial u_y}{\partial y} + \frac{\partial u_z}{\partial z} = 0 \tag{1}$$

with $(u_x, u_y, u_z)$ the Cartesian components of the ice velocity field.

The vertically integrated expression of the mass conservation equation (Eq. 1) provides the equation for the ice thickness, $H$:

$$\frac{\partial H}{\partial t} = -\frac{\partial H \bar{u}_x}{\partial x} - \frac{\partial H \bar{u}_y}{\partial y} + M - b_{melt} \tag{2}$$

with $\bar{u}_x$ and $\bar{u}_y$ the vertically integrated velocities in $x$ and $y$-directions, $M$ the surface mass balance and $b_{melt}$ is the basal melting rate.

The quasi-static approximation is used for the velocity field, in which the inertial terms of the momentum conservation equation are ignored. The gravity force being the sole external force acting on an infinitesimal cube of ice, we have:

$$
\quad
\begin{cases}
\dfrac{\partial \sigma_x}{\partial x} + \dfrac{\partial \tau_{xy}}{\partial y} + \dfrac{\partial \tau_{xz}}{\partial z} = 0 \\[3mm]
\dfrac{\partial \tau_{xy}}{\partial x} + \dfrac{\partial \sigma_y}{\partial y} + \dfrac{\partial \tau_{yz}}{\partial z} = 0 \\[3mm]
\dfrac{\partial \tau_{xz}}{\partial x} + \dfrac{\partial \tau_{yz}}{\partial y} + \dfrac{\partial \sigma_z}{\partial z} = \rho g
\end{cases}
\tag{3}
$$

where $\tau_{ij=x,y,z}$ are the shearing stress tensor terms and $\sigma_{i=x,y,z}$ the longitudinal stress tensor terms, defined as ($i = x, y, z$):

$$
\sigma_i = \tau_{ii} \tag{4}
$$

The pressure is defined as the first invariant of the stress tensor:

$$
-P = \frac{\sigma_x + \sigma_y + \sigma_z}{3} \tag{5}
$$

The deviatoric stress tensor is defined as (for $i, j = x, y, z$):

$$
\tau'_{ij} = \tau_{ij} + \delta_{ij} P \tag{6}
$$

with $\delta_{ij}$ being 1 for $i = j$, 0 otherwise.

Assuming isotropy, the deviatoric stress and the deformation rate $\dot{\epsilon}_{ij}$ are related by:

$$
\tau'_{ij} = 2\eta \dot{\varepsilon}_{ij} \tag{7}
$$

where $\eta$ is the ice viscosity.

Like most ice sheet models, GRISLI considers the ice as a non-Newtonian viscous fluid following a Norton-Hoff constitutive law (commonly named Glen flow law):

$$
\frac{1}{\eta} = B_{AT} \tau^{n-1} \tag{8}
$$

where $B_{AT}$ is a temperature-dependent coefficient and $\tau$ is the effective shear stress, defined as (for $i = x, y, z$):

$$
\quad \tau^2 = \frac{1}{2} \sum_{i,j} \tau_{ij}^2 \tag{9}
$$

The temperature-dependent coefficient $B_{AT}$ is computed following an Arrhenius' law:

$$
B_{AT} = S_f B_{BAT0} \exp\left( \frac{E_a}{R} \left( \frac{1}{T_m} - \frac{1}{T} \right) \right) \tag{10}
$$

where $S_f$ is a flow enhancement factor, $B_{BAT0}$ is a flow law coefficient, $E_a$ is the activation energy, $R$ the gas constant, $T_m$ the ice pressure melting temperature and $T$ the local temperature.

To account for the fact that the activation energy increases close to the melting point (Le Gac, 1980), in GRISLI, the pair ($E_a$, $B_{BAT0}$) can take two different values for local temperatures higher or lower than a temperature threshold $T^{trans}$ (Tab. 1).

The Glen flow law is an empirical formulation, derived from laboratory experiments. However, laboratory experiments cannot cover the full range of deviatoric stress operating in real ice sheets. The timescale over which this stress is applied in real ice sheets is also not reproducible in laboratories. Most modelling studies use $n = 3$ for the Glen flow law exponent (Eq. 8). A few studies suggest the possibility for a smaller exponent for small stress regime (Duval and Lliboutry, 1985; Pimienta, 1987; Pettit and Waddington, 2003). Since the work of Dumas (2002), one specificity of GRISLI is the possibility to simultaneously use a Glen viscosity with $n = 3$ and a linear, Newtonian, viscosity with $n = 1$. In this case, the two contributions simply add up:

$$2\,\dot{\varepsilon}_{ij} = \left(B_{AT1} + \tau^2 B_{AT3}\right)\,\tau'_{ij} \tag{11}$$

with $B_{AT1}$ and $B_{AT3}$ computed with Eq. 10, only using different activation energy ($E_{a\,1}$ and $E_{a\,3}$), flow law coefficient ($B_{BAT0\,1}$ and $B_{BAT\,3}$) and temperature threshold ($T_1^{trans}$ and $T_3^{trans}$). The chosen values (Tab. 1) are based on Lipenkov et al. (1997).

Like other large scale ice sheet models, GRISLI does not explicitly take into account anisotropy, which tends to facilitate deformation due to vertical shear, but reduces deformation due to longitudinal stress. The role of the flow enhancement factor $S_f$ in Eq. 10 is to artificially account for the effect of anisotropy. $S_f$ takes different values for the two components of the velocities computed in GRISLI (due to vertical shearing or longitudinal deformation, presented in the next section). In particular, we use a fixed ratio, typically ranging from 5:1 to 10:1 (Ma et al., 2010), between the value of the enhancement factor $S_f$ for vertical shear and the one for longitudinal deformation. In the following, this ratio will be set to 8:1 and the enhancement factor $S_f$ for vertical shearing with a Glen viscosity will be one of the calibrated parameters. In turn, the flow enhancement factor for the vertical shearing with a linear viscosity is set to 1.

### 2.1.2 Ice velocity

Differing from Ritz et al. (2001), the velocity in GRISLI is now computed for the entire domain as the superposition of the shallow ice approximation (SIA) and the shallow shelf approximation (SSA) components, without using a sliding law to estimate basal velocities.

The SIA (Hutter, 1982) assumes that the velocity horizontal derivatives are much smaller than its vertical derivatives, which is generally true for the major part of the ice sheet where the gravity driven flow induces a slow motion of the ice. GRISLI uses a zero order approximation of the SIA in which the stress tensor components simplify to:

$$\begin{cases} \tau_{xz} = \rho g \dfrac{\partial S}{\partial x}(S - z) \\[4mm] \tau_{yz} = \rho g \dfrac{\partial S}{\partial y}(S - z) \end{cases} \tag{12}$$

with $S$ the surface elevation. The vertical velocity profile is computed as an integral from the bedrock elevation, $B$, to a given vertical coordinate, $z$ (for $i = x, y$):

$$u_i(z) = u_{ib} + \int_B^z 2 \dot{\varepsilon}_{iz} \, dz \qquad (13)$$

where $u_{ib}$ is the $i$ component of the basal velocity.

The basal velocity can be computed with a sliding law (e.g. Bindschadler, 1983). However, recent versions of GRISLI use the shallow shelf approximation (SSA) as a sliding law as suggested by Bueler and Brown (2009). In this case, similarly to Winkelmann et al. (2011) we simply add up the two contributions of the SIA and SSA for the whole domain which ensure a smooth transition from non-sliding frozen regions to sliding over a thawed bed.

For fast flowing regions, the vertical stresses are much smaller than the longitudinal shear stresses. In this case, the velocity fields with the SSA (MacAyeal, 1989) reduce to the following elliptic equations:

$$
\begin{cases}
\dfrac{\partial}{\partial x} \left( 2 \bar{\eta} H \left( 2 \dfrac{\partial u_x}{\partial x} + \dfrac{\partial u_y}{\partial y} \right) \right) + \dfrac{\partial}{\partial y} \left( \bar{\eta} H \left( \dfrac{\partial u_x}{\partial y} + \dfrac{\partial u_y}{\partial x} \right) \right) = \rho g H \dfrac{\partial S}{\partial x} - \tau_{bx} \\[4mm]
\dfrac{\partial}{\partial y} \left( 2 \bar{\eta} H \left( 2 \dfrac{\partial u_y}{\partial y} + \dfrac{\partial u_x}{\partial x} \right) \right) + \dfrac{\partial}{\partial x} \left( \bar{\eta} H \left( \dfrac{\partial u_y}{\partial x} + \dfrac{\partial u_x}{\partial y} \right) \right) = \rho g H \dfrac{\partial S}{\partial y} - \tau_{by}
\end{cases}
\qquad (14)
$$

where $\tau_b$ is the basal drag. The velocities $u_x$ and $u_y$ are identical along the vertical dimension.

The condition at the front of the ice shelf is given by the balance between the water pressure and the horizontal longitudinal stress (see also Sec. 2.3 on the numerical features).

The code section relative to the elliptic equation is available in the supplement.

### 2.1.3 Basal drag

For floating ice shelves, the basal drag, $\tau_b$, is negligible. For cold-based grounded ice we impose a large enough basal drag (typically $10^5 \, \mathrm{Pa}$) to ensure virtually no-slip conditions on the bedrock and the basal velocity is set to zero in this case. For temperate-based grounded ice, a power-law basal friction (Weertman, 1957) is assumed:

$$
\begin{cases}
\tau_{bx}^m = -\beta \, u_{bx} \\[4mm]
\tau_{by}^m = -\beta \, u_{by}
\end{cases}
\qquad (15)
$$

where the basal drag coefficient $\beta$ is positive. In the experiment presented here, we assume the presence of a sediment at the base of the ice sheet allowing for a viscous deformation ($m = 1$).

In some recent applications of GRISLI, the basal drag coefficient has been inferred with an inverse method in order to match present-day ice sheet geometry (Ritz et al., 2015; Le clec'h et al., 2018). This approach has been followed to participate in the first phase of the recent ice sheet model intercomparison project (ISMIP6, Nowicki et al., 2016) for both the Greenland (Goelzer et al., 2017) and Antarctic ice sheets. In this context, GRISLI computes sea level rise projections by the end of the

century in line with results from more complex model (Edwards et al., 2014; Goelzer et al., 2017).

Inverse methods are especially suited to produce an ice sheet state (e.g. geometry and/or velocity) close to observations. However, by construction, such methods do not provide information where no ice is present in observations. As such they are difficult to apply for palaeo reconstructions of the American or Eurasian ice sheets. More generally, inverse methods are

no longer appropriate for long-term integrations, either palaeo or future, when ice thickness is very different from its present state and especially if the ice margin migrates from its present-day position. This motivates the use of a basal drag coefficient computed from GRISLI internal variables. We generally assume that its value is modulated by the effective water pressure, $N$, at the base of the ice sheet:

$$\beta = C_f N \tag{16}$$

with $C_f$ an internal parameter that needs calibration.

In our approach, any temperate-based grounded point will have a non-zero sliding velocity, depending on the $C_f$ factor used. Previous works have used additional criteria to limit the extension of these ice streams. For example, Alvarez-Solas et al. (2010) only compute Eq. 16 where the present-day sediment thickness exceeds a certain threshold whilst Quiquet et al. (2013) restrict this to large-scale valleys. However these approaches have flaws. For example, the sediment distribution is only poorly

known below present-day ice sheets and its past evolution is largely uncertain. Also, the definition of a typical spatial scale for ice streams is somewhat arbitrary. For these reasons, in the following we use the simplest approach and compute Eq. 16 for any temperate-based grid point. The $C_f$ parameter will be part of the calibrated parameters in our large ensemble.

### 2.1.4   Flux at the grounding line

In Ritz et al. (2001) the position of the grounding line was computed from a simple floatation criterion with no specific flux computation at the grounding line. Such an approach is in theory only valid for a very high spatial resolution, within tenths of meters, at the vicinity of the grounding line (Durand et al., 2009). Because the model runs typically at a much coarser resolution, in GRISLI version 2.0 we have implemented an explicit flux computation at the grounding line based on the analytical formulations from Schoof (2007) and Tsai et al. (2015). The two formulations differ from the assumption made on the sediment

rheology.

The flux at the grounding line following Schoof (2007) is:

$$q_{gl}^S = - \left( \frac{\bar{A}(\rho g)^{n+1}(1-\rho/\rho_w)^n}{4^n \beta} \right)^{\frac{m}{m+1}} H_{gl}^{\frac{m(n+3)+1}{m+1}} \phi_{bf}^{\frac{m\,n}{m+1}} \tag{17}$$

with $n$ and $m$ being the exponents in the Glen flow law (Eq. 8) and in the friction law (Eq. 15) respectively, $\bar{A}$ the vertically integrated temperature dependent coefficient in the Glen flow law ($B_{AT}$, Eq. 8), $H_{gl}$ the ice thickness at the grounding line, and $\phi_{bf}$ a back force coefficient to take into account the buttressing role of ice shelves. $\beta$ is the basal drag coefficient presented in Eq. 15.

Conversely, Tsai et al. (2015) proposed:

$$q_{gl}^T = Q_0 \frac{8\bar{A}(\rho g)^n}{4^n f} \left(1 - \rho/\rho_w\right)^{n-1} H_{gl}^{n+2} \phi_{bf}^{n-1} \tag{18}$$

with $Q_0 = 0.61$. In this case, the basal drag is assumed to vanish at the grounding and as such the coefficient $\beta$ is not used. Instead Tsai et al. (2015) suggests a constant and homogeneous basal friction coefficient $f$ set to 0.6.

In GRISLI, from the last grounded point in the direction of the flow, we compute the subgrid position of the grounding line in the $x$ and $y$ directions linearly interpolating the floatation criterion (dark green dots in Fig. 2). From this position, the flux at the grounding line is calculated using Eq. 17 or Eq. 18 (red arrows in Fig. 2). Because the flux at the sub-grid position is perpendicular to the local grounding line, ideally we should project this flux onto the x and y-axis. However, in the model, we assume that the grounding line is always perpendicular to either the x or y-axis (dashed brown line in Fig. 2). As in Fürst (2013), the value of the flux $q_{gl}$ is linearly interpolated to the two closest downstream and upstream velocity grid points (dark blue arrows in Fig. 2) using the two bounding velocity points (light blue arrows in Fig. 2).

To evaluate the back force coefficient $\phi_{bf}$, we solve the velocity equation twice. The first iteration is computed on the simulated geometry with no specific flux adjustment at the grounding line (i.e. not using Eq. 17 nor Eq. 18). The second iteration is computed the same way, except for the fact that the ice shelves are assigned to a very low viscosity so that they cannot exert any back force. The buttressing ratio $\phi_{bf}$ is then computed as the velocity ratio between these two computed velocities. Once $\phi_{bf}$ is estimated, we solve the velocity equation again, this time imposing the grounding line flux computation using Eq. 17 or Eq. 18, in order to estimate the velocity used in the mass conservation for this time step. We acknowledge the fact that this approach is computationally expensive but it allows for more accurate estimate for the buttressing role of ice shelves in the model.

The code for the implementation of the flux at the grounding line in GRISLI is available in the supplement.

### 2.1.5 Calving

Iceberg calving is not modelled explicitly. Instead, we used a simple ice thickness threshold criterion. Because this simple scheme can prevent ice shelf extension, we also maintain downstream ice shelf grid-points neighbouring the last grid-points meeting the criterion. The cut-off threshold may vary in space (e.g. oceanic depth dependency) and time. In the following, we use a constant and homogeneous thickness criterion (set to 250 m, roughly corresponding to the observed present-day Antarctic ice shelves front).

### 2.1.6 Ice temperature calculation

The ice temperature calculation has remained identical to Ritz et al. (2001). As in most large-scale ice sheet models (e.g. Winkelmann et al., 2011; Pollard and DeConto, 2012; Pattyn, 2017), the temperature in GRISLI is computed by solving the general advection-diffusion equation of temperature:

$$\frac{\partial T}{\partial t} = \underbrace{\frac{1}{\rho c}\frac{\partial}{\partial x}\left(k_i\frac{\partial T}{\partial x}\right) + \frac{1}{\rho c}\frac{\partial}{\partial y}\left(k_i\frac{\partial T}{\partial y}\right)}_{\text{horizontal diffusion}} \quad \underbrace{+ \frac{1}{\rho c}\frac{\partial}{\partial z}\left(k_i\frac{\partial T}{\partial z}\right)}_{\text{vertical diffusion}}$$

(19)

$$\underbrace{-u_x\frac{\partial T}{\partial x} - u_y\frac{\partial T}{\partial y}}_{\text{horizontal advection}} \quad \underbrace{-u_z\frac{\partial T}{\partial z}}_{\text{vertical advection}} \quad \underbrace{+\frac{Q}{\rho c}}_{\text{heat production}}$$

with $k_i$ the thermal conductivity of the ice, $c$ the heat capacity and $u_z$ the vertical velocity, computed as in Ritz et al. (1997) ($w_t$ in Eq. 14 of the original paper).

Horizontal diffusion is assumed to be negligible relative to the vertical diffusion.

The heat production is given by Hutter (1983) (for $i, j = x, y, z$):

$$Q = \sum_{i,j} \dot{\varepsilon}_{ij}\tau_{ij}$$
(20)

At the ice sheet surface, due to the absence of an explicit snowpack model, ice temperature is assumed to be equal to the near-surface air annual temperature (but not greater than the melting point). Depending on the surface mass balance parametrisation, the latent heat release due to refreezing is transferred to the first ice layer.

15 A geothermal heat flux $\phi_0$ is applied at the base of a 3 km thick bedrock layer with a Neumann boundary condition:

$$\phi_0 = -k_b\frac{\partial T}{\partial z}|_{bedrock}$$
(21)

with $k_b$ the bedrock thermal conductivity.

The heat equation is solved in the bedrock similarly to Eq. 19 but with no advection nor heat production. From the temperature gradient in the bedrock (computed on four vertical levels) we compute a heat flux $\phi_0'$ at the ice-bedrock interface. When ice

20 dragging occurs over the bedrock, an additional term due to friction, $\phi_f$, is added to $\phi_0'$:

$$\phi_f = |\mathbf{u}_b\, \tau_b|$$
(22)

The ice-bedrock interface heat flux is used differently for cold and temperate based points:

– For cold based points, the heat at the ice-bedrock interface is transferred to the ice via a Neumann boundary condition:

$$k_i\frac{\partial T}{\partial z}|_{ice} = -\phi_0' - \phi_f$$
(23)

with the ice thermal conductivity $k_i$ computed as:

$$k_i = 3.1014\ 10^8\ \exp(-0.0057\ (T + 273.15))$$ (24)

 – For temperate points, a Dirichlet boundary condition is applied as the temperature is kept at the pressure melting point. The excess heat in this case is used to compute basal melting:

$$b_{melt} = \frac{-\phi_0' - \phi_f - k_i \frac{\partial T}{\partial z}|_{ice}}{L_f \rho}$$ (25)

with $L_f$ is the latent heat of fusion.

Basal melting for oceanic points is usually imposed. For specific applications we have different values for deep ocean and continental shelves, or a geographical distribution depending on the oceanic basin.

The viscosity for the velocity grid points is the horizontal average of the viscosity on the centred grid and not the viscosity computed from the horizontal average of the temperature. This is preferable for regions with mixed frozen and temperate basal conditions.

## 2.2  Additional features

### 2.2.1 Basal hydrology

Peyaud (2006) added a simple diffusive basal hydrology scheme in GRISLI. In the following we provide a complete description of the hydrology model because it has only been described in a French Phd dissertation (Peyaud, 2006) but currently lacks a description in an international scientific journal.

Using a Darcy law, the water produced by melting at the base of the ice sheet is routed outside glaciated areas following the highest gradient in the total water potential.

Such a gradient can be written as in Shreve (1972):

$$\nabla\Phi = \rho_w g \nabla h_w + \rho_w g \nabla B + \rho g \nabla H \tag{26}$$

where $h_w$ is the hydraulic head, $B$ is the bedrock height and $H$ the ice thickness.

In GRISLI, we assume that the basal water flows within a sub-glacial till following a Darcy-type flow law:

$$\mathbf{Q_w} = -\frac{KD}{\rho_w g} \nabla\Phi \tag{27}$$

where $\mathbf{Q_w}$ is the water flux vector in $x$ and $y$ directions, $K$ is the hydraulic conductivity of the till and $D$ is the water thickness within the till.

The till is assumed to be present everywhere below the ice sheet with a constant and homogeneous thickness ($h_{till} = 20$m) 15   and porosity ($\phi_{till} = 0.5$). The water thickness in the till, $D$, equals the hydraulic head, $h_w$, only for thicknesses lower than the effective thickness:

$$D = \min(h_w, \phi_{till} h_{till}) \tag{28}$$

The hydraulic conductivity of the till, $K$, is modulated by the effective pressure to take into account sediment dilatation:

$$K = \begin{cases} K_0 & \text{if } N > N_0 \\ \\ K_0 \, N_0/N & \text{if } N \leq N_0 \end{cases} \tag{29}$$

with $K_0$ the reference conductivity, $N$ the effective pressure and $N_0$ a constant ($10^8$Pa). The conductivity $K_0$ is poorly constrained and strongly depends on the material.

In GRISLI, we assume that the flow of water within the till can be described with a diffusivity equation for the hydraulic head:

$$\frac{\partial h_w}{\partial t} + \nabla.\mathbf{Q_w} = b_{melt} - I_{gr} \tag{30}$$

with $I_{gr}$ being the infiltration rate in the bedrock (kept constant at $1 \, \text{mm yr}^{-1}$).

Using Eq. 26 and Eq. 27, this diffusivity equation can be written as:

$$\frac{\partial h_w}{\partial t} = b_{melt} - I_{gr} + \nabla.\left(KD\nabla\left(B + \frac{\rho}{\rho_w}H\right)\right) + \nabla.(KD\nabla h_w) \tag{31}$$

Eq. 31 can be solved using a semi-implicit relaxation method as for the mass conservation equation.

From the hydraulic head, $h_w$, we can compute the water pressure, $p_w = \rho_w \, g \, h_w$, and the effective pressure, $N = \rho \, g \, H - p_w$.

Fortran modules for the basal hydrology are available in the supplement.

### 2.2.2 Isostasy

As in Ritz et al. (2001), GRISLI computes the bedrock response to ice load with an elastic lithosphere - relaxed asthenosphere (ELRA) model (Le Meur and Huybrechts, 1996). This simple model evaluates the bedrock deformation to a local unit mass, scaled to the whole ice sheet. The relaxation time of the asthenosphere is usually set to 3000 years and the deflection of the lithosphere is assumed to follow a zero-order Kelvin function. Such a simple model has been shown to perform well compared to more sophisticated glacio-isostatic models (Le Meur and Huybrechts, 1996).

### 2.2.3 Passive tracer

GRISLI includes a passive tracer model that allows for the computation of vertical ice stratigraphy, i.e. time and location of ice deposition for the vertical model grid points. The model is the one of Lhomme et al. (2005) re-implemented in GRISLI by Quiquet et al. (2013). We use a semi-Lagrangian scheme following Clark and Mix (2002) in order to avoid the numerical instabilities of Eulerian schemes and information dispersion of Lagrangian schemes (Rybak and Huybrechts, 2003). For each timestep, the back trajectories of each grid points are computed and tri-linearly interpolated onto the model grid. This allows for a continuous information within the ice sheet at a low computational cost. Time and location of ice deposition can be convoluted for example with isotopic composition of precipitation (e.g. $\delta^{18}O$) in order to construct synthetic ice cores comparable with actual ice cores (Lhomme et al., 2005).

The GRISLI code section related to the passive tracers are available in the supplement.

### 2.3 Numerical features

The model uses finite differences computed on a staggered Arakawa C-grid in the horizontal plane (Fig. 2). In the vertical, the model defines $\sigma$-reduced coordinates, $\zeta = (S - z)/H$, for 21 evenly spaced vertical layers, with the $z$ vertical axis pointing upward and $\zeta$ pointing downward (0 at the surface and 1 at the bottom). The coordinate triplet $(i, j, k)$ (in $x$, $y$ and $\zeta$ direction) is representative of the centre of the grid cell. The horizontal resolution depends on the application, i.e. the extension of the geographical domain and the duration of the simulated period. For century scale applications, the resolution varies from 5 km for Greenland to 15 km for Antarctica (Peano et al., 2017; Ritz et al., 2015). For multi-millenial applications the resolution reduces to 15 km for Greenland and 40 km for the whole Northern Hemisphere and Antarctica.

The mass balance equation is solved as an advection-only equation with an upwind scheme in space and a semi-implicit scheme in time (velocities at the previous time step are used). The numerical resolution is performed with a point-relaxation method with a variable time step. The value of this time step is chosen to ensure that the matrix becomes strongly diagonal dominant to achieve convergence of the point-relaxation method. The criteria is thus a threshold that is inversely proportional to the fastest velocity on the whole grid. Note that this smaller time step is solely used for the mass conservation equation (Eq. 2) and subsequent variables (e.g. surface slopes, SIA velocity) while the rest of the model uses a main time step, typically

ranging from 0.5 to 5 years depending on the horizontal resolution.

To solve the ice shelves/ice streams equation, Eq. 14 needs to be linearised. The viscosity is computed using an iterative method starting from the viscosity calculated from strain rates from the previous time step. As this equation is the most ex-
pensive part of the model, the iteration mode is not always used depending on the type of experiment (for instance not crucial when the objective is to reach the steady state). In this case the viscosity of the previous time step is used. The linear system is solved with a direct method (Gaussian elimination, sgbsv in the Lapack library (www.netlib.org/lapack)).

The resolution of the elliptic system (Eq. 14) is the most expensive part of the model. This is further amplified by the way
we prescribe boundary conditions. As in Ritz et al. (2001), the ice shelf region is artificially extended towards the edges of the geographical domain. This artificial extension does not have any consequence on ice shelf velocity since added grid points (that we call "ghost"nodes) are prescribed with a negligible ice viscosity (1500 Pa s). The front is then parallel to either $x$ or $y$ (Ritz et al., 2001) and thus the boundary condition there is easy to implement (see also Fig. S1 in the supplement). The boundary condition at the real ice shelf front is solved implicitly with Eq. 14. However this method increases substantially the size of the
linear system solved in (Eq. 14). To circumvent this issue, a simple reduction method is implemented. (Eq. 14) can be written as $\tilde{A}\,\tilde{\mathbf{u}} = \tilde{B}$ where $\tilde{\mathbf{u}}$ is a vector alternating $u_x$ and $u_y$ components for all the velocity grid points, $\tilde{A}$ is a band matrix (very sparse) and $\tilde{B}$ is a vector corresponding to the right hand terms in Eq. 14. Every line of $\tilde{A}$ and $\tilde{B}$ are scaled so that the diagonal terms of $\tilde{A}$ are equal to 1. If, for a given velocity node, all the non diagonal terms of the column are very small compared to 1, this means that this node is in practice not used by any other velocity node and this line of the matrix can be removed. The
threshold to neglect nodes is related to the value of the integrated viscosity of "ghost" nodes . In practice, given its size, the matrix $\tilde{A}$ is not fully populated, being a band matrix with a bandwidth of one. An illustration of the matrix is shown in Fig. S2 in the supplement while the Fortran files are available ("New-remplimat" directory).

For the temperature equation (Eq. 19), we solved a 1D advection-diffusion equation for each model grid point. The resolu-
tion is performed with an upwind semi-implicit scheme (vertical velocity and heat production at the previous time step is used). The ice thermal conductivity is computed as the geometric mean of the two neighbouring conductivities (Patankar, 1980). Because the horizontal diffusion is neglected, the only horizontal terms concern horizontal advection and are computed with an upwind explicit scheme. The heat production is computed at the velocity (staggered) grid points and is then summed up to the temperature (centred) grid points.

The model has been recently partially parallelised with OpenMP (www.openmp.org), which considerably shortens the length of the simulations (gain of 40% for the Antarctic at 40 km on four threads of an Intel® Xeon® CPU@3.47 GHz).

## 3   Calibration for the Antarctic ice sheet

Over the years, several GRISLI internal parameters have been shown to be of importance to appropriately simulate the flow and mass balance of the Antarctic ice-sheet. Values for these parameters have been so far derived from non-systematic tests and expert knowledge. To systematically investigate the role of those parameters and find the best fitting set for the simulated

Antarctic ice-sheet with respect to the observed one, a calibration methodology with systematic exploration of the different values is performed in the following. The best fitting set will be considered as plausible models within the chosen parameter space. Given its degree of complexity, GRISLI is mostly designed for multi-millenial integrations. Due to long-term diffusive response to SMB and temperature changes, an accurate methodology to select unknown parameters of the model would be to run long transient simulations with a climate forcing as close as possible from past climate states, ideally with a synchronous

coupling between the ice sheet and the atmosphere. However, climate models generally fail at reproducing the regional climate changes during the last glacial-interglacial cycle as recorded by proxy data (Braconnot et al., 2012). Furthermore, the phase III of the Paleoclimate Modelling Intercomparison Project (PMIP3) has highlighted the large disagreement between participating climate models in simulating the Last Glacial Maximum (LGM) in the vicinity of northern Hemisphere ice sheets (e.g. Harrison et al., 2014). Given these uncertainties amongst climate models and the large sensitivity of the ice sheet model to climate forcing

fields (e.g. Charbit et al., 2007; Quiquet et al., 2012; Yan et al., 2013), it is difficult to calibrate the mechanical parameters independently from that of the SMB, in particular for northern Hemisphere ice sheets. For these reasons, here we suggest a simple calibration methodology for the Antarctic ice sheet in which the model is run for 100 kyrs under a perpetual modern climate forcing until equilibrium.

### 3.1   Methods

In the following, we use the 27 km-grid atmospheric outputs, namely annual mean temperature and SMB, from the regional climate model RACMO2.3 (Van Wessem et al., 2014), averaged over the 1976-2016 time span. The basal melting rates under ice shelves are prescribed for the 18 sectors of the Antarctic ice sheet as defined in ISMIP-Antarctica project (Nowicki et al., 2016) and are shown in Fig. 3. Their values are based on the sectoral average of sub-shelf melt rates that ensured stable ice shelves (minimal Eulerian ice thickness derivative) in the recent intercomparison exercise InitMIP-Antarctica (Nowicki et al.,

2016), with slight modifications due to change in resolution. They are in line with observations-based estimates (Rignot et al., 2013). We do not apply any correction related to geometry changes to the climatic forcings during the calibration.

We choose to restrict this study to a coarse horizontal resolution, namely 40 km, as it allows for large ensembles of multi-millenial simulations. Whilst 6.7 hours on one thread of an Intel® Xeon® CPU@3.47 GHz (4 hours on four threads) are

needed to perform 100 000 years of simulation over Antarctica on a 40-km grid (19 881 horizontal grid points), this time goes up significantly on a 16-km grid (145 161 points) for which we need 25 hours to perform 2000 years (17 hours on four threads). In addition, the 40 km resolution corresponds to the one used in the coupled version within the *i*LOVECLIM earth system model of intermediate complexity (Roche et al., 2014). Whilst with such a resolution we do not expect to have an ac-

curate representation of the ice sheet fine scale structures such as ice streams, we expect to reproduce the large scale behaviour of ice flow.

From our experience with GRISLI, we identified four unknown independent parameters that have a crucial role for ice dynamics:

- The SIA flow enhancement factor $S_f$ of the Glen flow law (Eq. 10). This coefficient is expected to have a large influence on shear-stress driven velocities.

- The basal drag coefficient $C_f$ in Eq. 16. This coefficient is used to modulate the basal drag coefficient for temperate-based grid points where sliding occurs.

- The till conductivity $K_0$. This parameter changes the efficiency of basal water routing (Eq. 29 and Eq. 31) and thus, basal effective pressure $N$. As such, this parameter is also influencing the basal drag coefficient $\beta$ for temperate-based regions (Eq. 16).

- An ice shelf basal melting rate coefficient $\phi_{shelf}$. For a specific Antarctic ice shelf sector $i$:

$$BMB^i = \phi_{shelf}BMB_0^i \tag{32}$$

with $BMB_0^i$ the sub-shelf basal melting rate reference values shown in Fig. 3.

The parametric ensemble is designed with a Latin Hypercube Sampling (LHS) methodology. The LHS is used here because it has better space-filling quality than a standard Monte-Carlo sampling which might not explore sufficiently the tails of parameter distributions. This methodology has been used for calibration purposes in the ice sheet modelling community (e.g. Stone et al., 2010; Applegate et al., 2012). The size of the LHS consists of 600 model realisations: half of it with the flux at the grounding line of Schoof (2007) (hereafter AN40S) and the rest with Tsai et al. (2015) (hereafter AN40T) The range of explored parameters are listed in Tab. 2. We assume an uniform statistical distribution within this range.

The initial ice sheet geometry, bedrock and ice thickness, is taken from the Bedmap2 dataset (Fretwell et al., 2013, Fig 4) using a spatial bi-linear interpolation to generate this data on the 40-km grid. The geothermal heat flux is from Shapiro and Ritzwoller (2004). Sensitivity to uncertainties in the forcing data are not explored in the ensemble as we aim at quantifying the model sensitivity to parameter choice even though we acknowledge for the fact that these could be the source of important model error (e.g. Stone et al., 2010; Pollard and DeConto, 2012).

In the following, individual member performance is assessed with the root mean square error (RMSE) computed from simulated and observed ice thickness (Fretwell et al., 2013) . This metric puts a strong constraints on ice sheet geometry and avoids potential compensatory biases that could appear when using total ice volume. Observed ice velocities are not used as a metric because of high spatial variability that results from small-scale bed properties. With a coarse 40-km resolution, the model is

intrinsically unable to reproduce such variability but is expected to reproduce the large scale pattern of ice thickness. However, at the end of the long 100-kyrs simulation, the velocities in the model are the balance velocities corresponding to the simulated topography. Thus, the minimisation of the RMSE in ice thickness should also reduce the error in velocities with respect to observations at the global scale.

## 3.2 Calibration results

Figure 5 presents the Antarctic ice sheet volumes at the end of the 100 kyr simulations for each ensemble member as a function of parameter values using the flux at the grounding line computed from Schoof (2007) (AN40S). We can see that there is a strong positive (respectively negative) correlation of ice volume with the basal drag coefficient (resp. enhancement factor). There is also a weak negative correlation for the sub-shelf basal melt coefficient, and a weak positive correlation with the till conductivity. Since the global volume is an integrated metric that does not account for potential systematic compensation, in Fig. 5 we also show the root mean square error (RMSE) in ice thickness for each ensemble members with respect to observations (Fretwell et al., 2013). Amongst the 300 model realisations, 120 members have a RMSE lower than 350 m. These members are widely distributed within the member spectrum. The lowest RMSE is 294 m. The 12 best ensemble members are outlined in red in Fig. 5 and have a RMSE not higher than 304.

The general model response is not fundamentally different when the flux at the grounding line is computed from Tsai et al. (2015) (Fig. 6). However, the grounding line position is much more unstable with a greater number of members showing lower ice sheet volume. Only 62 model realisations have a RMSE lower than 350 m (lowest RMSE at 295 m and 12 best not higher than 304), compared to 120 within the AN40S ensemble members. This difference in grounding line stability for the two flux formulations has already been highlighted by Pattyn (2017). Since basal drag vanishes at the grounding line in the Tsai et al. (2015) formulation, its coefficient has a smaller impact than in Schoof (2007), amplifying the role of the ice flow enhancement factor. As a consequence, for the AN40T ensemble, the enhancement factor requires values between 1.5 and 3 in order to reach a good agreement with observed ice thickness, whilst values within 1.5 to 4 are acceptable for AN40S.

In Fig. 7 (respectively Fig. 8) we show the ice thickness difference from the observations for the 12 ensemble members showing the lowest RMSE within the AN40S (resp. AN40T) model realisations. The differences are generally below 500 m even if persisting model biases are present across the ensemble members and model formulations. On the one hand, ice thickness in large parts of the East Antarctic ice sheet is systematically underestimated. On the other hand, the West Antarctic ice sheet shows more contrasted responses. Whilst for some ensemble members, the Ross embayment upstream region can be well represented (e.g. AN40S004 or AN40T065), the region feeding the Filchner-Ronne ice shelf show a quasi-systematic ice thickness underestimation. These model deficiencies can be attributed to our coarse model resolution, providing a poor representation of the complex bedrock structure in the Filchner-Ronne area. The model differences from the observations are very similar to results from the PISM-PIK model shown in Martin et al. (2011) in term of amplitude but also in term of structure. They are also generally similar to Pollard and DeConto (2009). Consistent model biases amongst these models, which use

different input data, suggest a common source of error related to the coarse model resolution (20 to 40 km) or uncertainties in the bedrock dataset, particularly large in East Antarctica (Fig. 4).

Figure 9 (resp. Fig. 10) presents the root mean square error with respect to observations of the ensemble members in the two-dimensional parametric space within AN40S (resp. AN40T). The 12 best ensemble members are outlined in red. In most cases there is no clear relationship. However, there is a relationship emerging with a large basal drag coefficient being compensated by a large enhancement factor when using the Schoof (2007) (Fig. 9). When using the Tsai et al. (2015) formulation ( Fig. 10), this relationship disappears as the enhancement factor is mostly driving the model response. A few model parameter combinations (30 out of 300) are able to provide a good representation of the present day Antarctic ice sheet, i.e. RMSE lower than 350 m, independently from the grounding flux computation used (not shown).

Although our quality metric is based on the ice sheet thickness we show in Fig 11 and in Fig. 12 the capability of the model to reproduce observed ice sheet velocity (Mouginot et al., 2017) for the best ensemble members with the two formulations of the flux at the grounding line (parameter values shown in Tab. 3). The model reproduces the general distribution of the velocity although it underestimates ice flow for the very fast grid points (velocity larger than $100 \; \mathrm{m \, yr^{-1}}$, generally ice shelves). The model has also difficulties in reproducing well-defined ice streams such as the Amery or Filchner ice shelves tributaries. Such difficulties could be due to the coarse resolution used but also to the simple scheme used to estimate basal drag.

From each of the two ensembles (AN40S and AN40T), we keep the 12 ensemble members out of 300 that have the lowest RMSE and use them in the next section for transient simulations covering the last 400 kyrs. Using these 24 plausible models on long term transient integration provides insight on the GRISLI result spread for models yielding a similar present-day ice sheet. Indeed, while they have a similar RMSE, they have distinct parameter values (Fig. 5 and Fig. 6) and as such they provide an insight of the uncertainties in ice sheet evolution relative to parameter choice. We acknowledge that the choice of 12x2 ensemble members is arbitrary and this number is too low to infer statistically meaningful results in terms of model uncertainty. Still, even with our relatively coarse resolution, 400 kyr-simulations represent a non-negligible computing time that has to be added to the 600 ensemble members of Sec. 3.

## 4   Antarctic ice sheet changes for the last 400 kyrs

### 4.1   Methods

By construction, equilibrium simulations such as the ones shown in Sec. 3 do not allow for the validation of the dynamical response of the flux at the grounding line since there are no climatic transitions and subsequent migration of the grounding line. The main objective of this section is thus to show the ability of the model to reproduce large ice sheet geometry changes in response to Quaternary climate change. As a consequence of our limited knowledge of past climatic conditions in the Antarctic ice sheet region over glacial-interglacial cycles, we use here an idealised reconstruction of SMB, near surface air temperature

and oceanic basal melting rates based on a limited number of proxy records. Our approach is somewhat similar to previous works (e.g. Ritz et al., 2001; Huybrechts, 2002; Pollard and DeConto, 2009; Greve et al., 2011; Golledge et al., 2014).

The near-surface air temperature, used in the model as a surface boundary condition for the advection-diffusion temperature equation, is assumed to follow the EPICA-DOMEC deuterium record $\delta D$:

$$T_{palaeo} = T_0 + \left(1/\alpha^i\right) \delta D \tag{33}$$

with $T_0$ the annual mean near-surface air temperature from RACMO2.3 used for the present-day calibration. The isotopic slope for temperature, $\alpha^i$, is set to $0.18\,\%o\,°\text{C}^{-1}$ as in (Jouzel et al., 2007).

We also account for the additional temperature perturbation due to topography changes using a fixed and homogeneous lapse rate $\lambda$:

$$T_{palaeo}^* = T_{palaeo} + \lambda\left(S - S_0\right) \tag{34}$$

with $S - S_0$ the local topography change from Bedmap2. In the following $\lambda$ is set to $-8\,°\text{C}\,\text{km}^{-1}$.

For a given near-surface air temperature change $T_{palaeo}^*$ relative to present-day $T_0$, we modify the present-day SMB field, $SMB_0$:

$$SMB = SMB_0\,\exp\left(-\gamma\left(T_0 - T_{palaeo}^*\right)\right) \tag{35}$$

with the precipitation ratio to temperature change $\gamma$ set to $0.07\,°\text{C}^{-1}$. The use of an exponential form in Eq. 35 is motivated by the Clausius-Clapeyron saturation vapour pressure for an ideal gas. Such a simple expression implies that SMB is driven only by accumulation, an assumption justified by the very little surface ablation experienced by the Antarctic ice sheet under present-day climatic conditions. However, we may underestimate the surface melt for warmer past interglacial periods.

In order to account for changes in basal melting rates below ice shelves, there is the need to define a continuous proxy covering several glacial-interglacial cycles for past sub-surface oceanic conditions around Antarctica. To this end, and due to the lack of such a record in the Southern ocean, we used the temperature derived from a benthic foraminifer $\delta^{18}O$ record from the North Atlantic. This temperature signal is considered to depict the North Atlantic Deep Water (NADW) temperature (Waelbroeck et al., 2002). Here, we assume that changes in NADW temperature drive changes in the temperature of waters upwelled in the Southern Ocean. This upward flow separates into surface equatorward and poleward flows, and thus influences surface and sub-surface temperature around coastal Antarctica (e.g. Ferrari et al., 2014). The basal melting rate below a specific ice shelf sector $i$, $BMB_{palaeo}^i$ for past periods is computed from its present-day value, $BMB_0^i$, corrected to account for past oceanic conditions:

$$BMB_{palaeo}^i = max\left(BMB_0^i\left(1 + \delta^{oc}\right),\,0.01\,\text{m}\,\text{yr}^{-1}\right) \tag{36}$$

using the palaeo-oceanic index $\delta^{oc}$ defined as:

$$\delta^{oc} = \alpha^{oc}\,\Delta T_{NA}/T_{NA0} \tag{37}$$

with $T_{NA0}$ the pre-industrial temperature deduced from North Atlantic benthic foraminifera (Waelbroeck et al., 2002) and $\Delta T_{NA}$ the deviation from this temperature in the past. $\alpha^{oc}$ is a conversion coefficient, set to 1 in the following.

The atmospheric and oceanic indexes, $T_{palaeo} - T_0$ and $\delta^{oc}$, used to drive the model for the last 400 kyr are presented in Fig. 13. In addition to these climatic perturbations we also use the eustatic sea level reconstruction of Waelbroeck et al. (2002) to account for sea level variations over glacial-interglacial cycles.

In the following, we discuss the model behaviour in response to the 400 kyr forcing. We performed simulations using the 12 parameter combinations from Sec. 3 that have the lowest RMSE for the two groups AN40S and AN40T, differing by the treatment of the flux at the grounding line. We used the 100-kyr integration under perpetual modern climate in Sec. 3 as a spin-up for the transient simulations. We further compute 10 kyr into the future with no climatic perturbation from the reference climate used in Sec. 3 in order to discuss the stability of the simulated present-day ice sheet state.

### 4.2 Transient simulation results

In Fig. 14 the simulated ice sheet volume is shown over the last 400 kyr. Across this time scale, a large glacial-interglacial volume variation is observed, in particular for the last two cycles where it reaches up to about 10 millions of $km^3$. In our simulations, the Antarctic ice sheet volume increase at the last glacial maximum (LGM, 21 kaBP) relative to pre-industrial corresponds to about -10 to -20 m of global eustatic sea level drop depending on the simulations. These numbers mostly fall in the range of previous ice sheet model reconstructions (e.g. Huybrechts, 2002; Philippon et al., 2006; Pollard and DeConto, 2009), Antarctic contributions inferred as the difference from far-field and Northern Hemisphere near-field estimates (Peltier, 2004) or near-field estimates (Ivins and James, 2005; Argus et al., 2014; Whitehouse et al., 2012; Briggs et al., 2014). Our reconstructions are nonetheless at the higher end of recent studies. This could be related to the fact that we do not account for different geologic bed types between today ice-free (with extensive amount of deformable till) and glaciated (mostly hard bed) continental shelf. To account for this, some authors have chosen a two-value basal drag for these different regions (e.g. Pollard and DeConto, 2012). Because of the large uncertainties related to the bed properties we have decided to ignore these differences, keeping in mind that this can bias our results towards thicker ice sheet when the ice expands over the continental shelf. In our simulations, the last interglacial (120 kaBP) ice volume has no substantial changes relative to the present-day ice volume, the Antarctic ice sheet contributing less than 6 centimetres to the global eustatic sea level rise in the simulations with the lowest RMSE at 0k. This is well below recent estimates, ranging from 3 to 7 m, inferred from the limited contribution of Greenland to the last interglacial highstand (Dutton et al., 2015). Our crude representation of the last interglacial climate in which no surface melt is possible may be the cause for such a discrepancy. In addition, our proxy-based basal melting rate does not allow for above than present basal melting rates during the last interglacial.

The uncertainty related to the choice of the internal parameters within our subset leads generally to up to 3 $10^6$ $km^3$ differences in our framework but does not change the model response to the forcings. In turn, the choice of either Schoof (2007) (AN40S) or Tsai et al. (2015) (AN40T) to compute the flux at the grounding line leads to important differences amongst tem-

poral model responses. AN40T systematically start to retreat before AN40S. It also produces a larger glacial to interglacial volume change. This confirms the fact that the Tsai et al. (2015) formulation leads to higher grounding line migration variability as already highlighted in Sec. 3 and by other authors (Pattyn, 2017). The additional 10 kyr into the future with no climatic perturbation shows that the AN40S ensemble members do not produce an ice sheet at equilibrium at 0 ka BP. This means that, in our model, the Schoof (2007) formulation produces unrealistically too slow post-LGM retreat which induces a model drift persisting till 10 kyr in the future. Conversely, the Tsai et al. (2015) formulation leads to more rapid retreat rates which provides a stabilisation of the ice sheet during the Holocene.

Simulated ice sheet surface elevations at selected snapshots for the two ensemble members with the lowest RMSE at 0 ka BP after the transient simulations are presented in Fig. 15. Ice sheet geometry during the last interglacial resembles the present-day one. This is particularly true for the eastern part whilst the West Antarctic ice sheet is only slightly thinner. At the last glacial maximum, the grounding line advances towards the edge of the continental shelf, in agreement with geological reconstructions (Bentley et al., 2014). The choice of the flux at the grounding line formulation has an impact on the maximum ice sheet extent, with a less extended ice sheet using the Schoof (2007) formulation. As for the last interglacial, the eastern part of the ice sheet presents only small variations in surface elevation compared to the present-day geometry. There is no decrease in surface elevation at the last glacial maximum due to reduction in precipitation at this time since the larger extent and the colder climate tend to reduce the ice flow. The largest topography changes are occurring in the Weddel and Ross sea. The West Antarctic ice sheet is thus particularly dynamic during glacial-interglacial cycles.

The RMSE computed at 0 ka BP for the 24 members used for the transient simulations ranges from 372 to 467 m within AN40S and 326 to 376 m within AN40T. These numbers are only slightly greater than the ones obtained using a constant forcing (Sec. 3). The differences in ice sheet thickness between the transient simulations at 0 ka BP and the observations for the two ensemble members with the lowest RMSE (AN40S097 and AN40T059) are shown in Fig. 16. The pattern is similar to the one obtained during the calibration step (Sec. 3) with some notable differences. On the one hand, the East Antarctic ice sheet thickness underestimation is partly corrected when performing a transient simulation. This could be the result of a better representation of the temperature vertical profile in this case. On the other hand, whilst in other regions the model biases remain generally the same between an equilibrium and a transient simulation, important model biases appear at the margins of the ice sheet when using the transient simulations. This is particularly visible when using Schoof (2007) for which the retreat rate during the deglaciation is underestimated. Part of the misrepresentation of present-day margins could also be due to the over-simplified climatic perturbation used for the transient simulations.

## 5 Discussion and outlook

We have presented results from the updated version of the GRISLI model. Whilst the model is able to reproduce present-day Greenland (Le clec'h et al., 2017) and Antarctic (Ritz et al., 2015) ice sheets when using an inverse method to estimate the basal

drag, our simulations with an interactive basal drag coefficients computed from the effective pressure show some important disagreements relative to observations. In particular there are some persisting model biases in ice thickness. In East Antarctica, the ice thickness is underestimated towards the pole and the Transantarctic mountains while it is overestimated towards the margins, from Queen Maud land to Wilkes land. In West Antarctica, there is an underestimation of ice thickness in the Ronnie-Filchner basin and an overestimation in the Ross basin. These model biases are also present in models of similar complexity when using an interactive basal drag computation (Pollard and DeConto, 2009; Martin et al., 2011). This data-model mismatch is mostly due to a poor representation of the bedrock-ice interface. In particular, the coarse resolution does not allow for the consideration of fine scale troughs and pinning points. The persisting model biases can also be the consequence of our simplified basal drag computation that does not take into account bedrock physical properties (e.g. sediments).

We used a basal drag coefficient computed from an internal model parameter, namely the basal effective pressure. For long-term multi-millenial integrations, this is preferred to deducing the basal drag coefficient from inversion using present-day geometry since it is fully consistent with the model physics and, in principle, remains valid for large ice sheet geometry change. However, by design, the fit with observations is systematically poorer compared to model results that make use of an inverse basal drag coefficient. A step forward would be to use the basal drag computed from inversion in order to deduce a formulation based solely on internal parameters. Amongst these parameters, along with the basal effective pressure, the large scale bedrock curvature and/or sub-grid roughness could be used, as in Briggs et al. (2013). However, some key basal features, such as the geologic bed type and the deformable till distribution, remain today largely unknown below present-day ice sheets and will contribute to large uncertainties in the basal drag formulation.

Although widely used for ice sheet model spin-up or calibration, long-term integrations under present-day forcing induce a warm bias in the vertical temperature profile because they discard the diffusion of glacial-interglacial changes in surface temperature. Calibrated parameters obtained with such a methodology tend to compensate for the under-estimated viscosity and are in theory not suitable for palaeo-reconstructions. Whilst a parameter calibration based on glacial-interglacial simulations is ideally preferred, the determination of a realistic climate forcing is a considerable challenge given the many degrees of freedom. Here, we presented a very simplified climate reconstructions for the last 400 kyr based on a minimal parameter set (proxy for atmospheric temperatures and oceanic conditions) in order to illustrate the model possible behaviour for long-term integrations. Using the parameters calibrated under perpetual modern climate, the model is nonetheless able to reproduce ice geometry changes compatible with palaeo-constraints. Further work will consist in the determination of more realistic climate reconstruction using general circulation model snapshots. We also aim at expanding the work of Roche et al. (2014) and couple the Antarctic geometry of GRISLI version 2.0 with the *i*LOVECLIM model.

The implementation of an explicit flux computation at the grounding line following Schoof (2007) and Tsai et al. (2015) lead to a more dynamic grounding line position compared to previous version of the model. As such, GRISLI version 2.0 is now more sensitive to both sub-shelf melt rate changes and also sea level variations. However, the current version of the model only con-

siders an eustatic sea level perturbation with a regional bedrock adjustment. The explicit computation of local relative sea level could potentially have an important impact on grounding line migration for glacial-interglacial cycles (e.g. Gomez et al., 2013).

In the current version of the model, some important processes are still largely simplified. In particular, further developments will consist in the implementation of a new basal hydrology model relying on an explicit routing scheme (e.g. Kavanagh and Tarasov, 2017) avoiding relaxed numerical solutions based on effective pressure. This could introduce fast basal water changes that are currently ignored and, ultimately, could yield ice streams abrupt speed-up or slow-down. Also, calving processes are suspected to be a major driver for ice sheet evolution due to the importance of buttressing on inland ice dynamics (e.g. Pollard et al., 2015). GRISLI version 2.0 includes a very simplified calving representation that might prevent to assess the role of this process for multi-millenial ice dynamics. The inclusion of a physically based calving scheme (e.g. Christmann et al., 2016) would be a significant model improvement for future model revisions.

## 6   Conclusions

We have presented the GRISLI (version 2.0) model along with the significant improvements from the previous version of Ritz et al. (2001). Such improvements include an explicit flux computation at the grounding line, an interactive basal hydrology module and a semi-lagrangian tracking particle scheme. Thanks to its low computational cost, the model is suitable for long-term multi-millenial integrations. We performed a large ensemble of simulations of the Antarctic ice sheet forced by present-day climate conditions to calibrate the crucial unknown parameters. We have shown that the model is able to reproduce reasonably well the present-day geometry although the grounding line position in the model is much more unstable when we use Tsai et al. (2015) formulation of the flux at the grounding line instead of Schoof (2007). The model mismatch with respect to observed ice thickness shows some systematic biases (e.g. the East Antarctic ice sheet is too thick in the vicinity of the Transantarctic mountains and too thin elsewhere), that are similar to models of comparable complexity. We used the best ensemble members to simulate the Antarctic evolution throughout the last 400 kyr using an idealised climatic perturbation of present-day conditions. With this simple framework we reproduced the expected ice sheet geometry changes over glacial-interglacial cycles. A significant volume increase is simulated during glacial periods with a grounding line advance towards the edge of the continental shelf. The retreat during terminations is gradual when using our forcing scenario and is able to produce a final present-day ice volume and extent similar to observations. The Tsai et al. (2015) formulation produces a faster ice sheet retreat and yields an ice sheet near equilibrium during the Holocene contrary to Schoof (2007) for which the model is still drifting at +10 kyr into the future. This suggests that, in our model and under the climate forcing scenario we use, the Tsai et al. (2015) formulation produces a more realistic grounding line retreat rate.

## 7 Code availability

The developments on the GRISLI source code are hosted at https://forge.ipsl.jussieu.fr/grisli. It is at present in a transitional phase with the aim at releasing it publicly in the future but is currently not publicly available. Access to the full code can be granted on demand by request to Christophe Dumas (christophe.dumas@lsce.ipsl.fr), Aurélien Quiquet (aurelien.quiquet@lsce.ipsl.fr) or Catherine Ritz (catherine.ritz@univ-grenoble-alpes.fr) to those who conduct research in collaboration with the GRISLI users group. For this work we used the model at revision 188. Sections of the code used in the current manuscript that are currently under the CeCILL licence are made available as a supplement to this paper. Provided files include: the resolution of the elliptic equation, the implementation of Schoof (2007) and Tsai et al. (2015), the basal hydrology and the passive tracer.

*Acknowledgements.* We thank Michiel van den Broeke (IMAU, Utrecht University) for providing the RACMO2.3 model outputs. We also warmly thank Claire Waelbroeck for fruitful discussions on the construction of the index for sub-shelf melting rates. This is a contribution to ERC project ACCLIMATE; the research leading to these results has received funding from the European Research Council under the European Union's Seventh Framework Programme (FP7/2007-2013)/ERC grant agreement 339108.

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

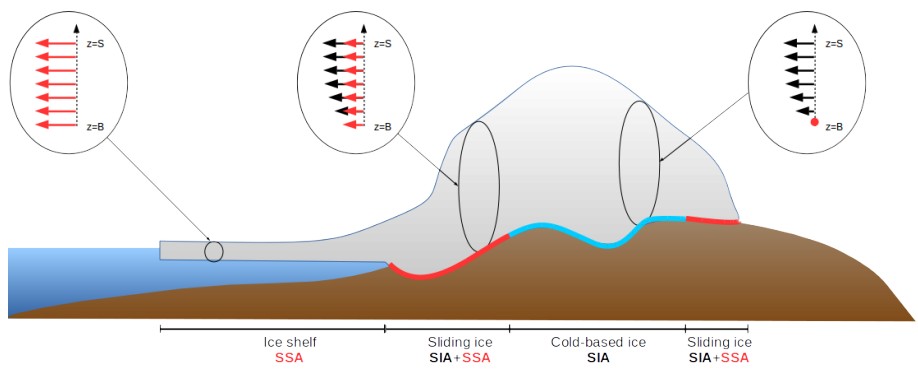

**Figure 1.** Schematic representation of the different types of flows in GRISLI and their associated velocity profiles. The red arrows stand for the sliding velocity which is non-zero for temperate-based grounded regions.

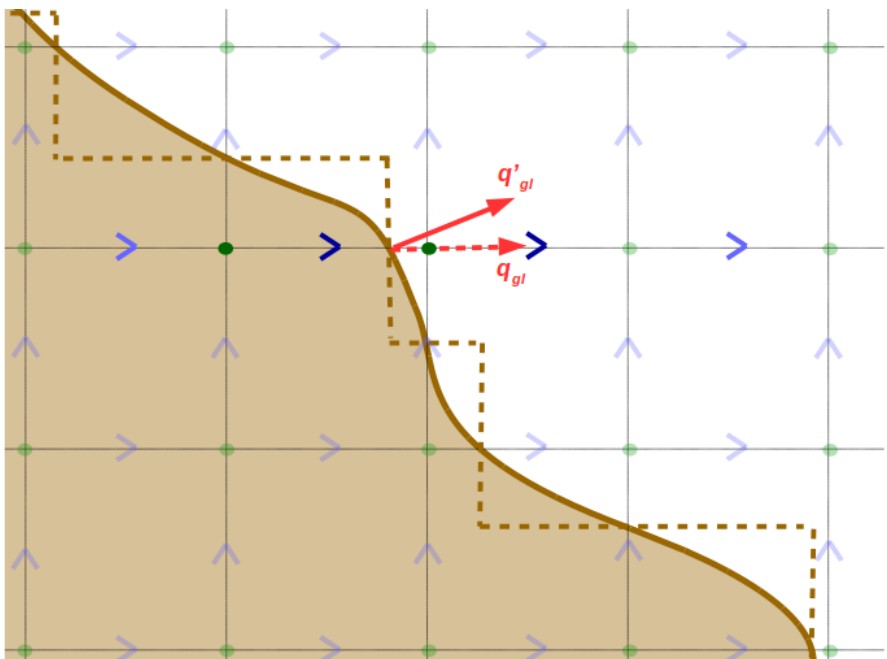

**Figure 2.** Horizontal staggered grids used in the model. The blue arrows stand for the staggered velocity grid while the green circles represent the standard centred grid (for, e.g., ice thickness, temperature, effective pressure). The plain brown line is an illustration of the grounding line position with an example of the flux ($q'_{gl}$) at one location where the grounding line crosses the x-axis in the centred grid. In the model, the norm of the flux at this location is reported on the $u$ velocity only ($q_{gl}$), i.e. assuming a grounding line perpendicular to the x-axis (dashed brown line). The dark blue arrows are the velocity nodes on which $q_{gl}$ is interpolated onto, using the velocity values of the two bounding light blue arrows. The dark green dots are used to infer the sub-grid position of the grounding, interpolating the floatation criterion ($\rho H + \rho_w (B - sealevel)$).

**Table 1.** GRISLI model parameters used in this study.

| Variable | Identifier name | Value |
|---|---|---|
| Global constants | | |
| Gravitational acceleration | $g$ | $9.81\,\mathrm{m\,s^{-2}}$ |
| Water density, liquid | $\rho_w$ | $1000\,\mathrm{kg\,m^{-3}}$ |
| Water density, ice | $\rho$ | $918\,\mathrm{kg\,m^{-3}}$ |
| Water density, ocean | $\rho_o$ | $1028\,\mathrm{kg\,m^{-3}}$ |
| Isostasy | | |
| Mantle density | $\rho_m$ | $3300\,\mathrm{kg\,m^{-3}}$ |
| Relaxation time in isostasy | $\tau_m$ | $3000\,\mathrm{yr}$ |
| Lithosphere flexural rigidity | $D_l$ | $9.87 \times 10^{24}\,\mathrm{N\,m}$ |
| Radius of relative stiffness | $R_l$ | $131910\,\mathrm{m}$ |
| Radius of action of a unit mass | $R_{iso}$ | $400\,\mathrm{km}$ |
| Hydrology | | |
| Thickness of the till layer | $h_{till}$ | $20\,\mathrm{m}$ |
| Porosity of the till layer | $\phi_{till}$ | $0.5$ |
| Deformation | | |
| Transition temperature of deformation, Glen | $T_3^{\mathrm{trans}}$ | $-6.5\,^{\circ}\mathrm{C}$ |
| Activation energy below transition, Glen | $E_{a\,3}^{\mathrm{cold}}$ | $7.820 \times 10^4\,\mathrm{J\,mol^{-1}}$ |
| Activation energy above transition, Glen | $E_{a\,3}^{\mathrm{warm}}$ | $9.545 \times 10^4\,\mathrm{J\,mol^{-1}}$ |
| Flow law coefficient below transition, Glen | $B_{AT0\,3}^{\mathrm{cold}}$ | $1.660 \times 10^{-16}\,\mathrm{Pa^{-3}\,yr^{-1}}$ |
| Flow law coefficient above transition, Glen | $B_{AT0\,3}^{\mathrm{warm}}$ | $2.000 \times 10^{-16}\,\mathrm{Pa^{-3}\,yr^{-1}}$ |
| Transition temperature of deformation, linear | $T_1^{\mathrm{trans}}$ | $-10\,^{\circ}\mathrm{C}$ |
| Activation energy below transition, linear | $E_{a\,1}^{\mathrm{cold}}$ | $4.0 \times 10^4\,\mathrm{J\,mol^{-1}}$ |
| Activation energy above transition, linear | $E_{a\,1}^{\mathrm{warm}}$ | $6.0 \times 10^4\,\mathrm{J\,mol^{-1}}$ |
| Flow law coefficient below transition, linear | $B_{AT0\,1}^{\mathrm{cold}}$ | $8.373 \times 10^{-8}\,\mathrm{Pa^{-3}\,yr^{-1}}$ |
| Flow law coefficient above transition, linear | $B_{AT0\,1}^{\mathrm{warm}}$ | $8.373 \times 10^{-8}\,\mathrm{Pa^{-3}\,yr^{-1}}$ |
| Temperature | | |
| Gas constant | $R$ | $8.314\,\mathrm{J\,mol^{-1}\,K^{-1}}$ |
| Ice melting temperature, at depth $z$ | $T_m$ | $9.35 \times 10^{-5}\rho g(S-z)\,\mathrm{K\,kPa^{-1}}$ |
| Latent heat of ice fusion | $L_f$ | $335 \times 10^3\,\mathrm{J\,kg^{-1}}$ |
| Ice thermal conductivity, at temperature T | $k_i$ | $3.1014 \times 10^8 \exp\left(-0.0057\left(T+273.15\right)\right)\,\mathrm{J\,m^{-1}K^{-1}yr^{-1}}$ |
| Mantle thermal conductivity | $k_b$ | $1.04 \times 10^8\,\mathrm{J\,m^{-1}K^{-1}yr^{-1}}$ |

**Table 2.** Selected parameters included in the latin hypercube sampling (LHS) ensemble with their associated ranges.

| Parameter | Units | Minimum | Maximum |
|---|---|---|---|
| $S_f$ | – | 1. | 5. |
| $C_f$ | $\mathrm{yr\,m^{-1}}$ | $0.5\ 10^{-3}$ | $5.\ 10^{-3}$ |
| $K_0$ | $\mathrm{m\,yr^{-1}}$ | $20.10^{-6}$ | $200.10^{-6}$ |
| $b_{melt}$ | – | 0.75 | 1.25 |

**Table 3.** Parameter values for the ensemble members that yield the lowest RMSE with respect to observations at the end of 100-kyr simulation under perpetual present-day climate forcing.

| | ensemble member | $S_f$ | $C_f$ ($\mathrm{yr\,m^{-1}}$) | $K_0$ ($\mathrm{m\,yr^{-1}}$) | $\phi_{shelf}$ |
|---|---|---|---|---|---|
| Using Schoof (2007) | 123 | 3.19 | $2.4\ 10^{-3}$ | $188\ 10^{-6}$ | 1.05 |
| Using Tsai et al. (2015) | 213 | 2.33 | $4.6\ 10^{-3}$ | $114\ 10^{-6}$ | 0.86 |

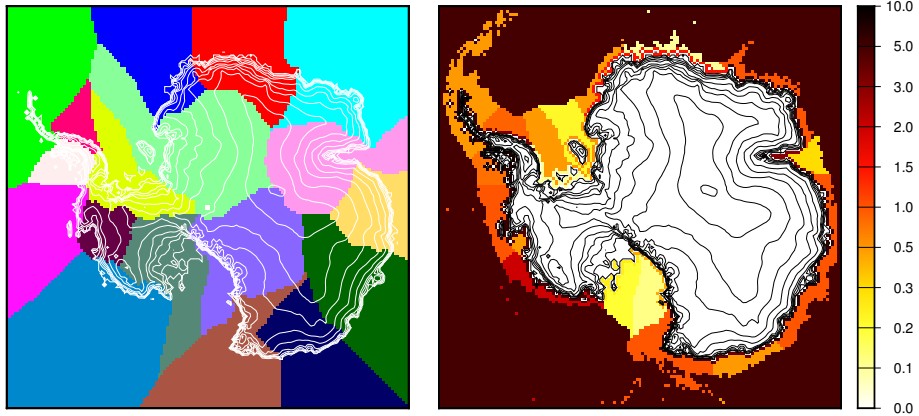

**Figure 3.** Antarctic ice shelves sectors (left) and associated prescribed present-day sub-shelf basal melting rates in $\mathrm{m\,yr^{-1}}$ (right). The melting rates are different for the shelf and the associated grounding line to mimic the higher values observed close to the grounding line (Rignot et al., 2013). Sub-shelf melting rate for the deep ocean (depth greater than 2500 m) are assigned a value of $5\,\mathrm{m\,yr^{-1}}$.

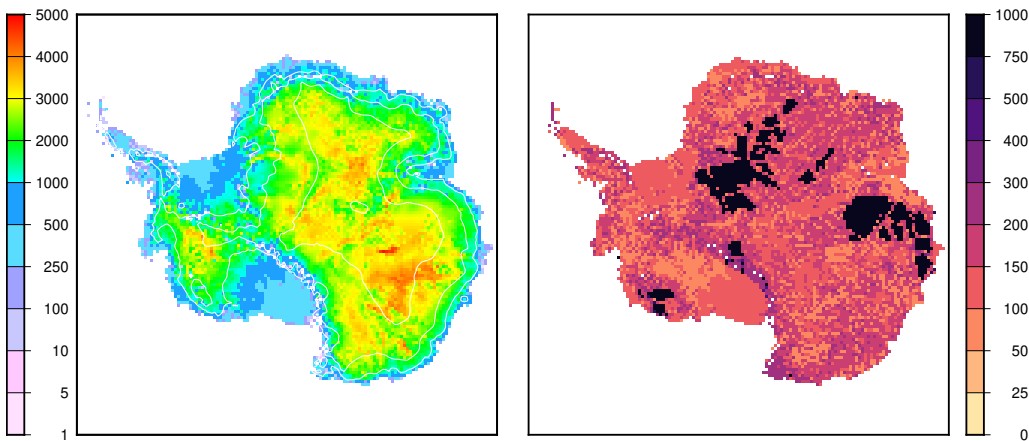

**Figure 4.** Bedmap2 ice thickness (left) and associated uncertainty (right) in metre (Fretwell et al., 2013), interpolated on the GRISLI grid of Antarctica at 40 km resolution. Despite considerable improvements from Bedmap1, large areas present an important uncertainty ($\pm$ 1000 m) due essentially to a poor in-situ data coverage.

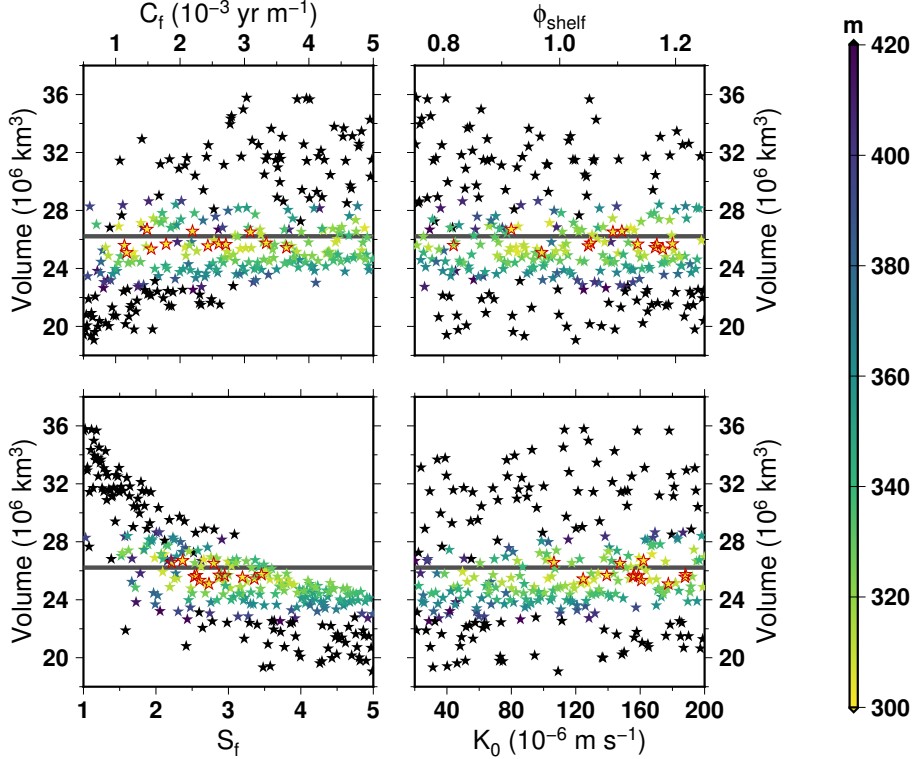

**Figure 5.** Simulated total ice volume for each ensemble members as a function of parameter values when using the Schoof (2007) formulation of the flux at the grounding line (AN40S). The thick horizontal line shows the observations (Fretwell et al., 2013). The colour shading corresponds to the root mean square error in ice thickness relative to observations. The stars outlined in red are the 12 ensemble members having the lowest RMSE.

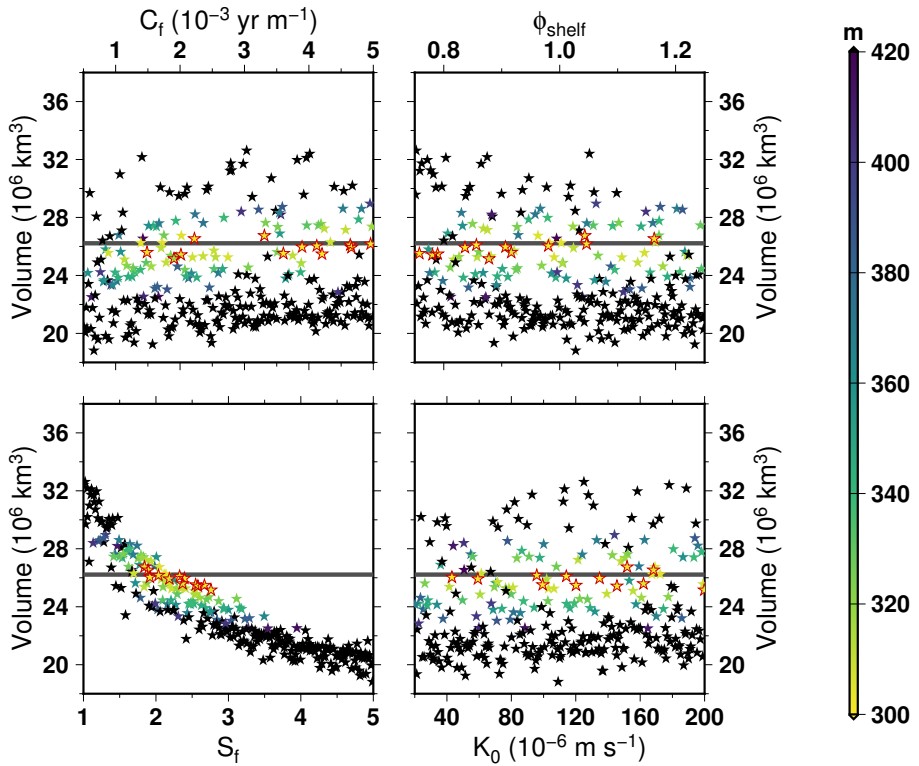

**Figure 6.** Same as Fig. 5 but with the Tsai et al. (2015) formulation of the flux at the grounding line (AN40T).

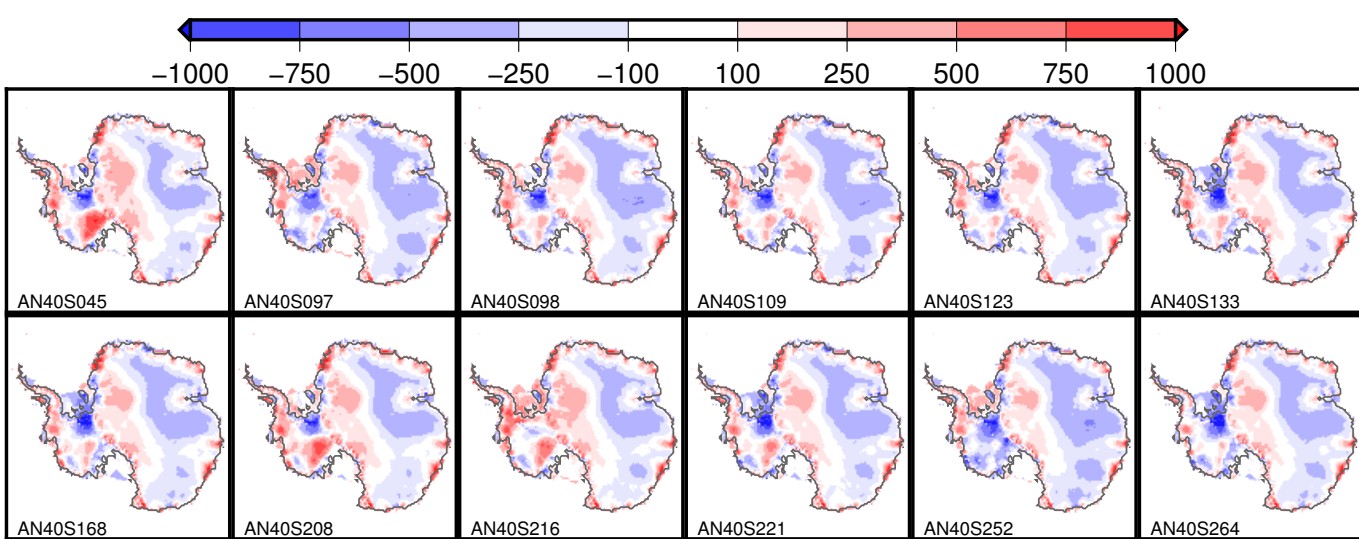

**Figure 7.** Ice thickness difference with the observations in metres (simulated minus observed) from the 12 ensemble members showing the lowest RMSE when using the Schoof (2007) formulation of the flux at the grounding line (AN40S).

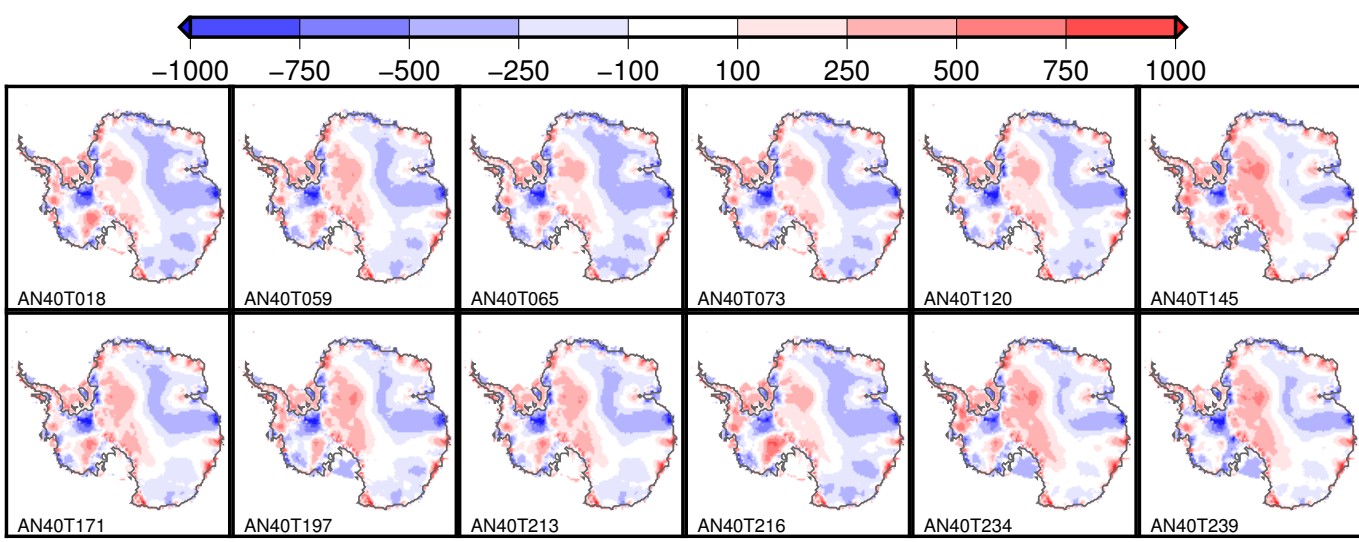

**Figure 8.** Same as Fig. 7 but with the Tsai et al. (2015) formulation of the flux at the grounding line (AN40T).

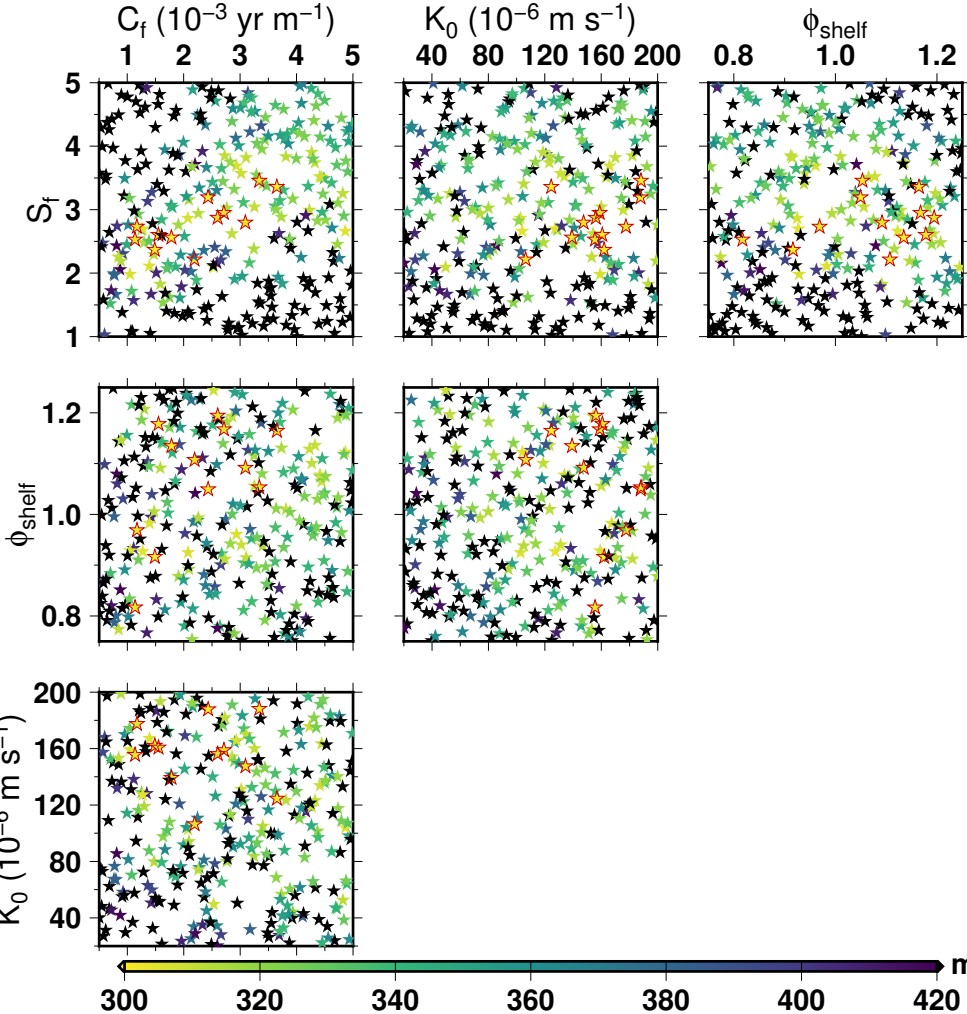

**Figure 9.** Ice thickness root mean square error respective to observations in the parameter space for the 300 model members using the Schoof (2007) formulation of the flux at the grounding line (AN40S). The 12 experiments showing the lowest error are outlined in red.

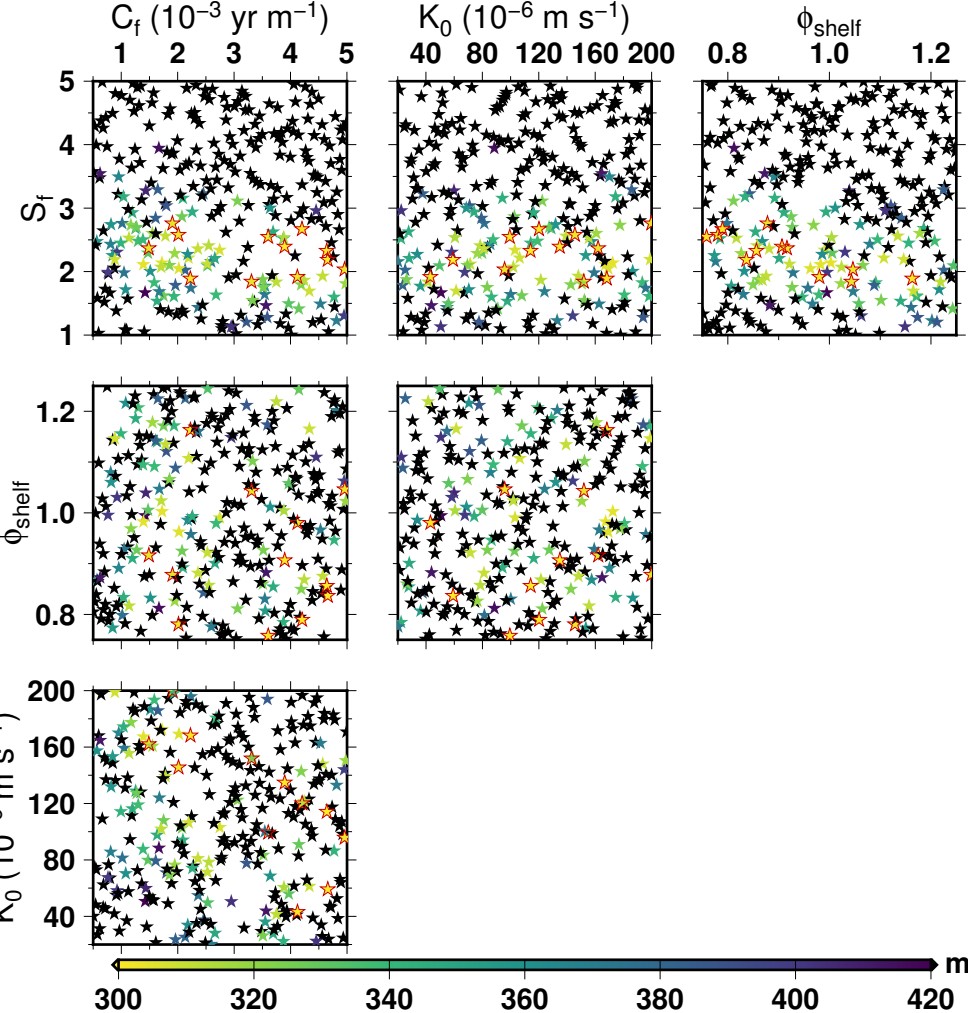

**Figure 10.** Same as Fig. 9 but with the Tsai et al. (2015) formulation of the flux at the grounding line (AN40T).

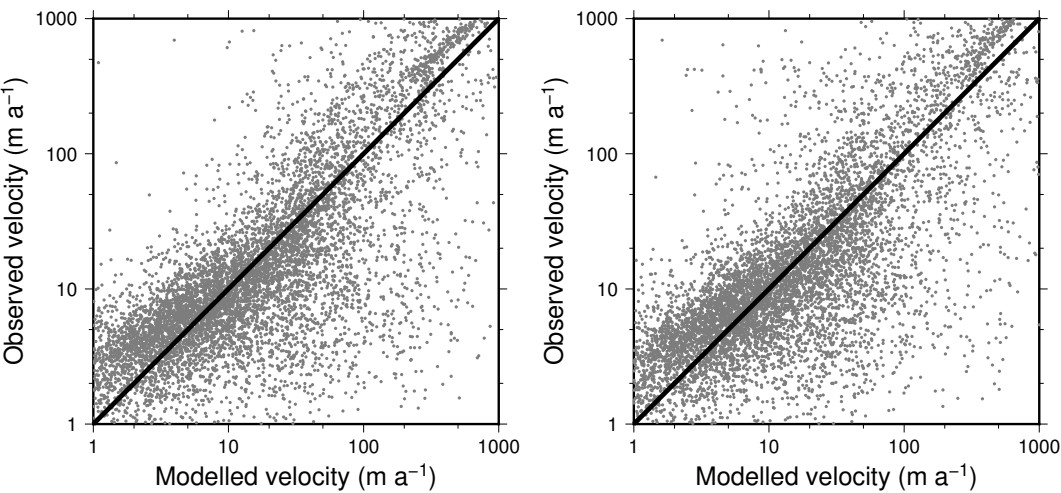

**Figure 11.** Observed velocity (Mouginot et al., 2017) against modelled velocity on the 40 km grid. Only the best ensemble members with the lowest RMSE is shown here for Schoof (2007) (AN40S123, left) and Tsai et al. (2015) (AN40T213, right). The parameter values for these experiments are shown in Tab. 3.

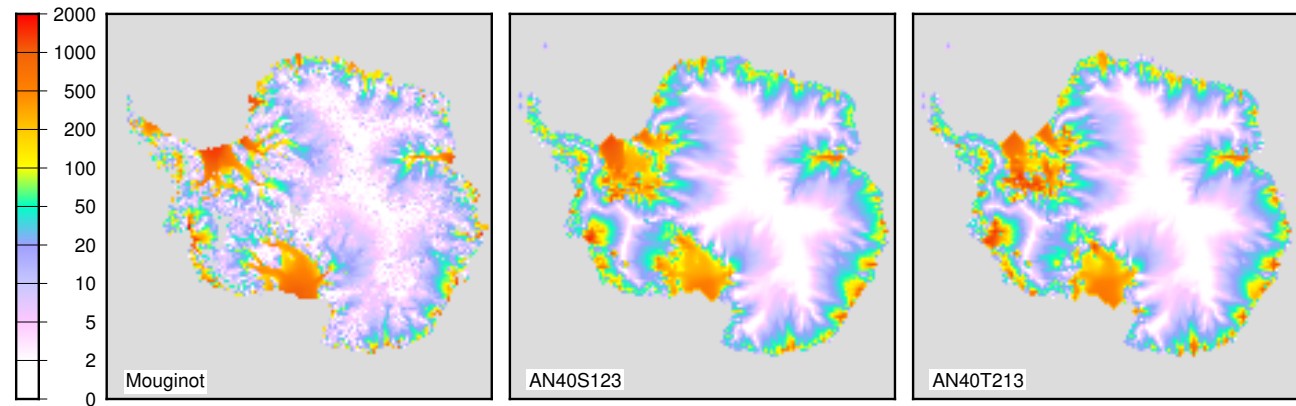

**Figure 12.** Map of observed (Mouginot et al., 2017) and simulated velocities in $\mathrm{m\,yr^{-1}}$ for the ensemble members with the lowest RMSE using Schoof (2007) (AN40S123) and Tsai et al. (2015) (AN40T213).

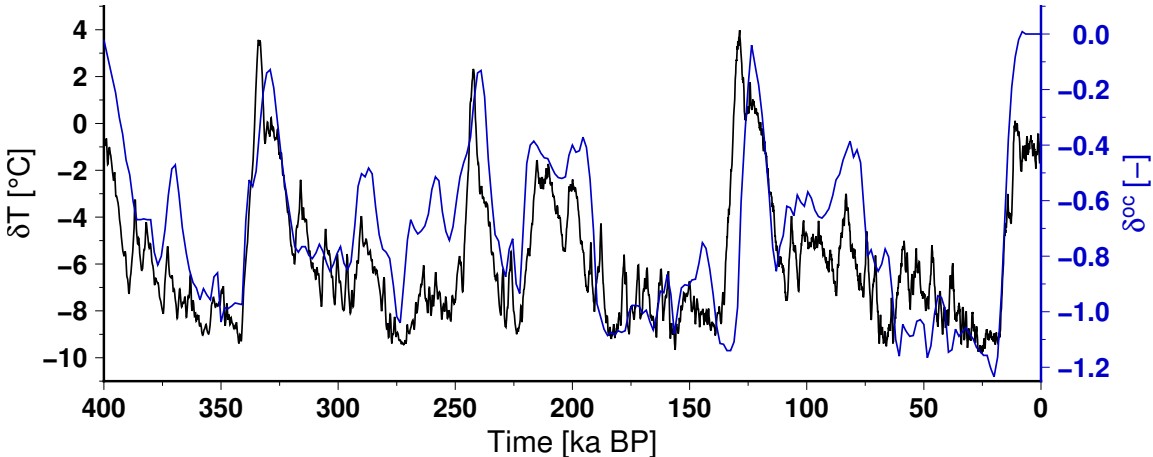

**Figure 13.** Climatic perturbation used in the 400 kyr glacial-interglacial simulations for the near-surface air temperature, $\delta T = \left(1/\alpha^i\right)\delta D$, and for the sub-shelf basal melting rate modificator, $\delta^{oc} = \alpha^{oc}\,\Delta T_{NA}/T_{NA0}$.

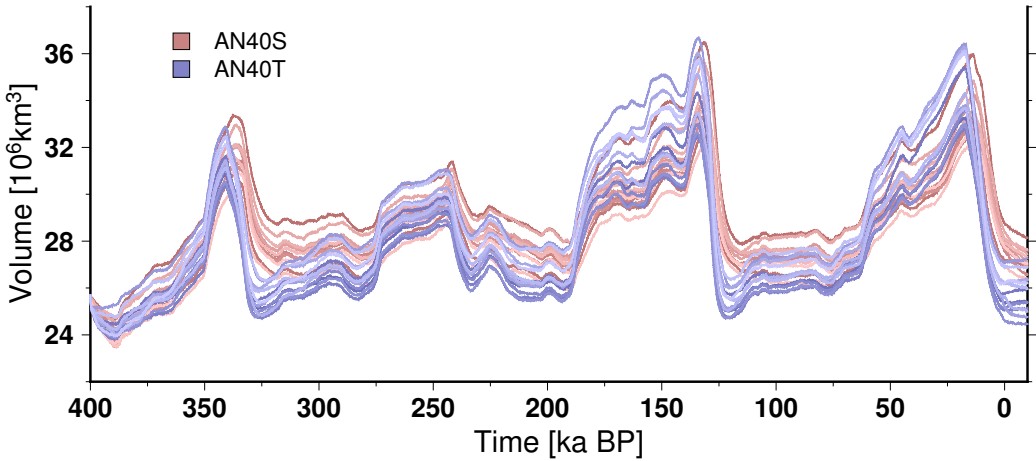

**Figure 14.** Simulated total ice sheet volume evolution over the last 400 kyr for the twelve ensemble members showing the lowest RMSE in Sec. 3 when using the flux at the grounding line computed from Schoof (2007) (AN40S, shade of reds) and Tsai et al. (2015) (AN40T, shade of blues). The glacial to interglacial difference in ice volume for the last termination corresponds to about -10 to -20 m of global sea level rise equivalent depending on the simulations.

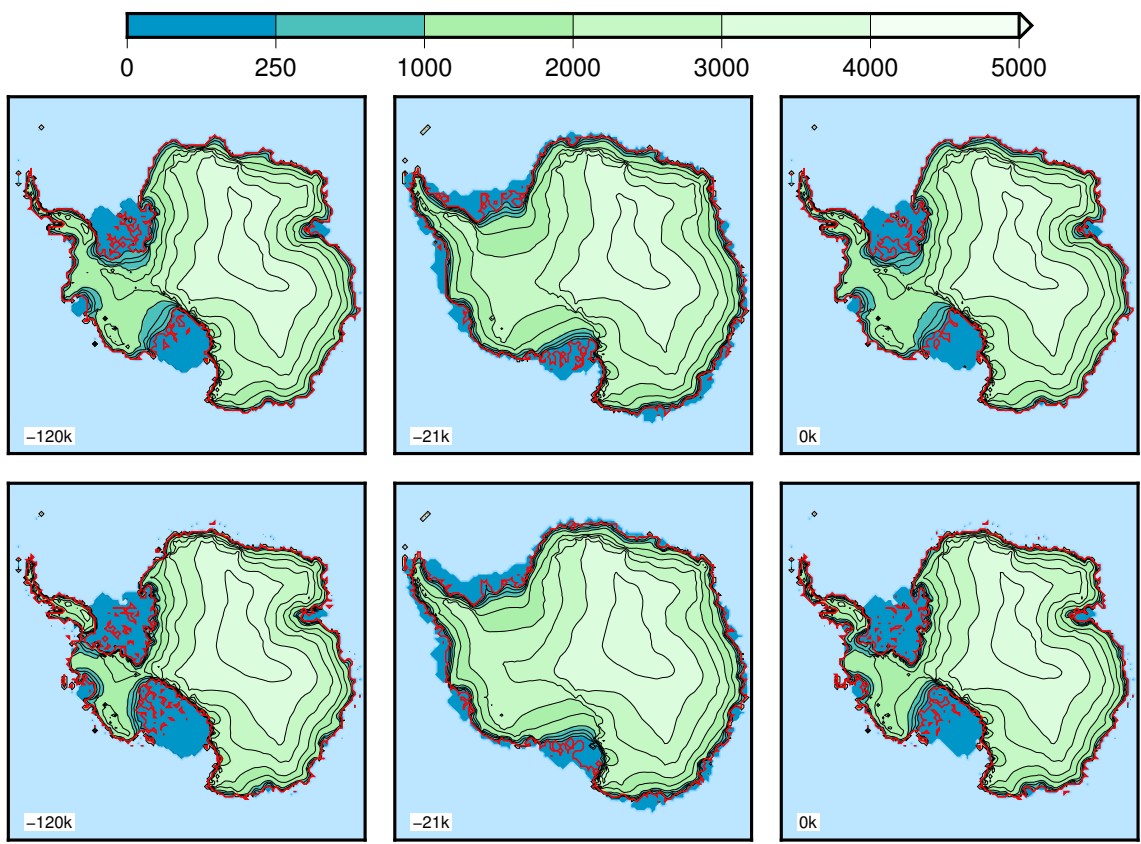

**Figure 15.** Simulated surface elevation at selected snapshots for the two ensemble members that produce the minimal RMSE at 0 kaBP in the transient simulations (AN40S252, top, and AN40T213, bottom). The ice volume contributing to sea level change from present is -9.3 m (resp. -15.1 m) at 21 kaBP for AN40S252 (resp. AN40T213), whilst it is limited at 120 kaBP (-0.3 m and +0.6 m). The grounding line is indicated by the thick red line.

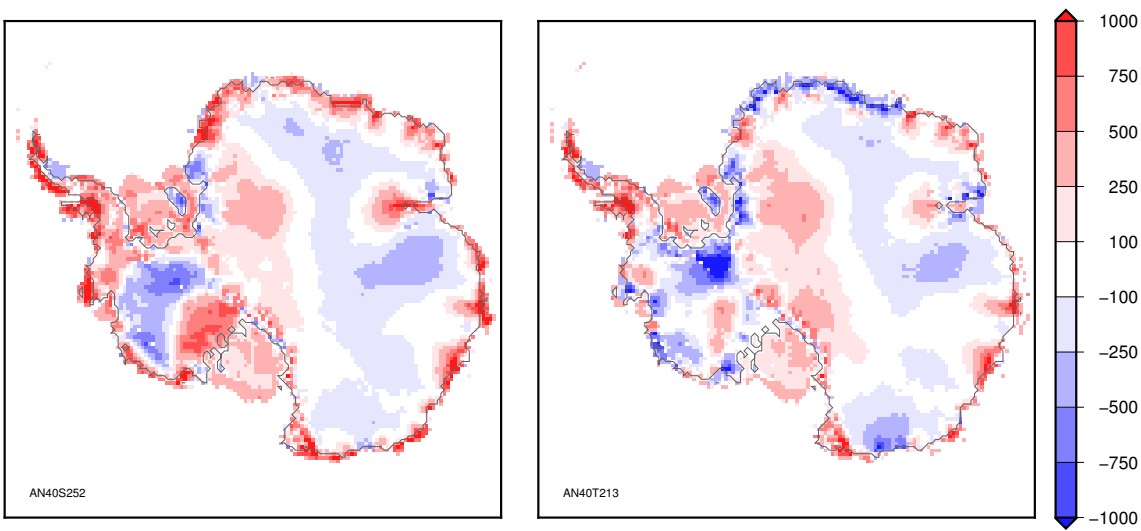

**Figure 16.** Ice thickness difference with the observations (simulated minus observed) at 0 ka BP for the two ensemble members that produce the minimal RMSE at 0 kaBP in the transient simulations (AN40S252, left, and AN40T213, right). The RMSE is 350 m (respectively 313  m) for AN40S252 (resp. AN40T213).