# Peer review of "The GRISLI ice sheet model (version 2.0): calibration and validation for multi-millennial changes of the Antarctic ice sheet"

_Geoscientific Model Development, 2018_

## Referee Comment (RC1) · Anonymous Referee #1 · 17 May 2018

Overall:

This paper provides a description of the current GRISL1 ice sheet model, focusing on modifications and extensions from the previously described version in 2001. The model is designed for relatively coarse-resolution long-term paleo applications. Main change from 2001 include the specification of grounding-line fluxes (Schoof, 2007), and a basal hydrology model. A large ensemble with Latin Hypercube sampling is used to constrain and calibrate four uncertain model parameters. The paper is clear and provides useful supporting documentation for GRISL1 users and background for other papers on GRISL1 applications.

[Figure]

Specific comments:

1. There is considerable scatter in Fig. 7, showing pair-wise parameter correlations with the results (here, rms error in modern ice thickness). This may be because the Latin Hypercube (LHC) sampling of the large ensemble (LE) may be too coarse to meaningfully detect pair-wise dependencies. The quasi-random distribution of red, blue and green stars in these panels is reminiscent of corresponding figures in Applegate et al. (2012, The Cryo., their Fig. 1). Chang et al. (GMD, 2014) subsequently found that the scatter in the Applegate study is due to inadequate sampling in the high-dimensional parameter space, and they used additional statistical analysis with Gaussian emulation to extract meaningful dependencies (their Fig. 4a vs. 4b). That study had a similar number of parameters (5) and ensemble members (100) as here (4 and 150). In a similarly sized Antarctic LE, Pollard et al. (GMDD, 2016) found that meaningful dependencies could only be found with "full-factorial" sampling, i.e., a run for every possible combination of parameter values, requiring 5ˆ4 = 625 runs for 4 parameters and 5 values each. If that many runs could be performed here, it might yield much more meaningful pair-wise results than in Fig. 7. However, if that would be too computationally expensive, it could be left to future work, and the above caveats could just be noted.

2. The simulations here use uniformly prescribed basal drag coefficients, and do not use an inversion method to deduce a spatial map. There is discussion on the pros and cons (pg. 6, 15, 17), which makes good points for not using inverse methods. But it does not mention the primary motivation (I think) for using them: that without them, modern errors in ice thickness are much larger (as in Figs. 5,6), and can be made much smaller using an inverse procedure. Since these errors are the primary metric here for evaluating the model, this could help to make the calibration of model parameters more meaningful. I think the whole issue depends on whether the inverse-produced map captures real bed variations at all, or if it just cancels with and obscures other physical errors in the model. I suggest mentioning this within the existing discussion. Also, the

point made on pg. 17, line 16, on the desirability of making basal coefficients a property solely of internal model parameters (such as N here, in Eq. 14), is debatable: apart from basal temperatures and water amount of course, spatial variations in basal sliding can also depend importantly on geologic bed type, roughness, and the distribution of deformable till, which are outside the scope of the model.

3. In Eq. 14 on pg. 6, and section 2.2.1 (Eqs. 24-26, pg. 9), it is not clear how some variables for basal hydrology are determined: $h_w$ or $p_w$ (which are related, line 24), and effective pressure N needed for Eq. 14. Presumably there is a prognostic equation in the hydrology model for $h_w$, i.e., $d(h_w)/dt = ...$, that is not shown here. Perhaps it is the equation mentioned on pg. 9, line 23. Also N possibly depends on $p_w$. This information, and the equation for $h_w$, should be included. (Incidentally, if N depends on depth below sea level as in several other models, I would question how can it reasonably depend on that, at distances 10's or 100's km inland from the grounding line).

4. The determination of buttressing factor $\phi_{bf}$ in Eq. 15 (pg. 7) is an important part of the use of the Schoof flux equation, but the procedure is unclear to me from lines 21-25 on that page. Perhaps the first solution provides the back-stress-free solution...does that solution use Eq. 15 with $\phi_{bf} = 1$? Then what is the second solution, and where does its value of $\phi_{bf}$ come from? These questions may not make sense, and just show my confusion. Hopefully this paragraph can be clarified, and perhaps expanded if that would help.

5. pg. 16, lines 5 to 7: Perhaps, the timings of the deglacial retreat in AN40T vs. AN40S can be assessed vs. papers in the RAISED reconstruction volume (Bentley et al., 2014), or other data, in order to determine which one is more realistic. The paper seems to decide rather arbitrarily that the AN40T case is more realistic (pg. 18, line 25).

6. pg. 13, lines 9-10: The pairs of values "1.5 to 3" and "1.5 to 5" do not seem to relate

to the bottom-right panels in Figs. 3 and 4, for basal-drag coefficients K0 (which are being discussed in that sentence). They seem to relate better (but still fuzzily) to the bottom-left panels for enhancement factors E_sia.

Technical comments:

The English usage is generally good, but isolated words or phrases could be improved/corrected, some of which are noted below.

pg. 1, line 19: Change "are evidences" to "is evidence".

pg. 1, line 23: Change "An other" to "Another".

pg. 2, line 7: I think "prograde" should still be "retrograde", for MICI as well as for MISI.

pg. 2, line 23: The word "diffusion" should probably be removed (?).

pg. 3, line 14, and several later places: "Tab. 1" should perhaps be "Table 1".

pg. 4, line 1: The use of two "respectively"'s in the same sentence is confusing - perhaps divide into 2 separate statements for sigma_i and tau_ij.

pg. 4, line 26: Change "Alike" to "Like".

pg. 4, lines 27-29: The word "reduces" in line 27 seems to contradict the word "favour" in line 29. Or perhaps "longitudinal" should be "shearing" in line 29 (?).

pg. 5, line 24: What does "see also numerical feature" mean?

pg. 16: I would suggest emphasizing, as a positive note, that if the (interpolated) grounding line position is known, then all that is required to obtain ice thickness H_gl at the grounding line for Eq. 15 is (1) bedrock bathymetry interpolated to the grounding line position, and (2) sea level. (This is because of the floatation criterion at the grounding line of course).

pg. 10, line 23-24: Explain the need for the artificial extension (to get an ice front parallel to x or y).

pg. 11, line 20: Misspelled "projet".

pg. 12, line 15: "in 150" should be "of 150".

pg. 12, line 20: Perhaps change "are discarded from" to "are not included in" ?

pg. 14, line 6: Change "somehow" to "somewhat".

pg. 15, line 16: Does this mean that a 100-kyr long spinup is performed with perpetual modern climate for every ensemble member? If so, say that more clearly.

pg. 15, line 24: The range of 10 to 20 m eustatic sea level drop here is actually a bit larger than several recent model studies. This might be due to larger basal drag coefficients used here on modern continental shelves, so when grounded ice expands onto them at LGM, the expanded ice is thicker there.

pg. 16, line 4: Change "In turns" to "In turn".

pg. 17, line 2: "using an inverse method" sounds like one is used here. Make it clear that one is not, and that phrase refers just to the references earlier in the sentence.

Table 1: Some of these variables do not seem to be used in the text, e.g., those under "Deformation". Others may have a different name, e.g., h_till.

Fig. 2: The relationship between the sector boundaries (left-hand panel) and the contour divisions for basal melt rates (right-hand panel) is confusing, not as one might expect. That is, there seems to be some divisions between the colors in the right-hand panel that are not present in the left-hand panel, and vice-versa.

---

## Referee Comment (RC2) · F. SAITO (Referee) · 4 Jun 2018

This paper describes the numerical ice-sheet model GRISLI ver 2.0, in particular, with the application for the Antarctic ice sheet simulation. I think this paper is fairly well written with some exception below, and can be accepted with minor revision.

One thing better to include is technical (numerical) procedures and properties adopted in the model. Since the source code is not opened to public, information how to solve the model equation is useful for those who has not contact with the model. For example, how to solve the Eq.(2), (thickness evolution)? Explicit, implicit or others? How to solve the linear equations, direct, alternate-direction, conjugate gradient, or other

method? How to solve the non-linear ice-shelf equation (Eq.12)? Linearize them? Or the velocity-dependent viscosity ($\bar{\eta}$) at the previous time step is used? How to determine the convergence of the solutions where the iterative solver is involved? Such details are all necessary to evaluate the numerical accuracy of the model, if they want. I suppose they are more or less the same as the original version of the model (Ritz et al. 1997,2001), but repetition (or at least citation of the old papers) are needed for completeness of the model description.

Another thing better to rewrite is the lateral boundary condition of the ice shelf, which is still unclear to me. As described in Sec 2.3, the ice shelf is extended towards the edges of the model domain. As far as I understand, to remove row i from the matrix corresponds to set horizontal velocity as zero at the front. I am not sure this is what the authors expect. I suggest to rewrite the second paragraph of the Sec.2.3 to clarify how to formulate the matrix in the model.

In addition, I definitely agree to the specific comments 3 and 4 by the referee #1. The authors should clarify the formulation and the procedures to compute basal hydrology and the back-force coefficient.

Some minor points (PmLn corresponds to the line number n in page m)

Units: use \unit{} macro.

P1L6: 'or Tsai et al....' may be better?

P2L16: 'right' might not be a right word for this context. How about 'practical tool for....'

P3L19 Eq(1): Divergence, not Gradient (need dot).

P3L22 Eq(2) and after: $\bar{u}_x$ is better than $\bar{u_x}$.

P3L26 Eq(3) and similar array equations: Use \displaystyle.

P4L1 or around: Need definition like $\sigma_i = \tau_{ii}$, otherwise the paper misses the equation for longitudinal stress components, since Eqs 5, 6, 8, 9 are described with

$\tau_{ij}$.

P4L11 Eq(7) or (9): The enhancement factor should be inserted. Otherwise $E_{SIA}$ in P12L1 is confusing.

P4L16 Eq(8): Need range of i,j.

P4L25 Eq(9): No explicit formulation of $B_{AT1}$ and $B_{AT3}$. Are they documented in Dumas (2002)?

P5L10 Eq(10): i in $\rho_i$ conflicts with row i.

P5L9: (for i = x,y) not (for i=x,y,z). When i=z in Eq(11), the coefficient 2 must disappear.

P5L10 Eq(11): no definition of B. B may conflict with $B_{AT}$, which is better to avoid.

P5L21: S is already defined in L9.

P5L23: (see also 2.3 numerical feature)

P5L26: The basal drag is very small but not necessarily zero. All we can do is to neglect it.

P7L12 Eq(16): No definition of $H_g$. Typo of $H_{gl}$?

P8 Sec2.1.6. Need to mention how to compute the vertical velocity. I suppose vertical velocity is not directly computed as Ritz et al. (1997).

P8L10: 'zero' requires the unit. I prefer to write as 'the melting point'.

P8L25 Eq(22) Write \exp (backslash before exp) following LaTeX convention.

P9L27, Better to cite Le Meur and Huybrechts after ELRA sentence also.

P10L26: Better to avoid to use A and B for matrix and vector, which conflict with the rate factor or bedrock elevation.

Table 1: Unit of the acceleration should be m/s. Really same values for ice and mantle

thermal conductivity?

Figure 1: If z at the ice bottom is always zero as the figure, you need to reformulate all the governing equations using z.

SAITO Fuyuki.

---

## Referee Comment (RC3) · Anonymous Referee #3 · 21 Jun 2018

Review of "The GRISLI ice sheet model (version 2.0): calibration and validation for multi-millennial changes of the Antarctic ice sheet" by Quiquet et al.

Summary:

This paper describes a new version of the GRISLI ice sheet model, including new model development since an earlier version many years ago (Ritz et al., 2001). While there are no major scientific advances relative to the state-of-the-art manifested with this update, documenting the current state of the model and the individual progress that has been made is well appreciated. The paper is generally well written and clear. Nevertheless, I believe there could be more detail given in some of the descriptions to

make it a better reference and make the paper more accessible for other modellers. I consequently recommend publication in GMD with some corrections as detailed below.

General comments:

I believe it is a good practice to (regularly) publish model description papers like the present one, to document the applied models, increase transparency and allow for other modellers to learn, improve upon and critically evaluate the applied techniques. One point I find regrettable with the present paper is that the authors seem to not plan to publish the model code alongside with the manuscript. I know that this may not be common practice in our community, but I believe it would be an important step forward. I would applaud if the authors would think about how to make the code publicly available, possibly with certain restrictions.

While the applied modelling techniques are mostly well described in words and equations (textbook style), the numerical implementation is often not possible to determine. I invite the authors to make an additional effort to increase the precision. This is even more important when the model code cannot be consulted. The ultimate goal should be that someone who does not know the model would be able to implement a specific feature from the given information. See also specific points below.

Reference to textbooks and earlier works is mostly in order. However, in some cases it would be useful for the reader to have some additional ('meta') information for the specific descriptions. E.g. who else is using the same technique, or does the applied technique represent a notable difference/novelty compared to other models used in the community. If other approaches exist, do you have reasons to choose this approach compared to another and why (simplicity, better results, tried other approach but didn't work ...). Again, the motivation should be to make the paper a useful reference and interesting resource for another modeller trying to implement a specific feature.

Specific comments:

P1 L14 Which time scales or time scale range, be more specific.

P1 L15 "surface albedo" is not a feedback, nor "freshwater flux". Please clarify.

P1 L18 Not sure about the connection between the two sentences implied here.

P1 L21 I would suggest rewording to avoid drastic terms "rapid" and "destabilisation".

P1 L22 "The surface mass balance-height feedback has ..."

P2 L1-3 Not all bedrock in the Antarctic shows a retrograde slope. More precision needed.

P2 L4 MISI driven retreat does not have to be very fast. Suggest removing "fast and"

P2 L7 Add "ice" before "cliff"

P2 L7 Buttressing already decreases when the ice shelve is removed and a reason for ice cliffs to fail. Reformulate.

P2 L10 What is the range of temporal and spatial scales, specify.

P2 l10 "temporal and spatial scales" of what exactly?

P2 L14 There are also state of the art models that are not FS, clarify.

P2 L19 Remove "Conversely".

P2 L21 add "e.g." before Hindmarsh

P2 L23 Reformulate "temperature and surface mass balance perturbations diffusion"

P2 L26 New sentence with "GRISLI was in the late nineties ..."

P3 L4 Not only MISI, but also GL movement in general. Reformulate.

P3 L15 "2.1 Ice thermo-mechanics". I didn't find the "thermo" aspect in this section. Reword?
P3 L21 Add equation number after "mass conservation equation".

P3 L23 "the vertically integrated velocities in x and y-direction u_x and u_y".

P4 L1 "where sigma ...  and tau ...  are the longitudinal and shearing stress tensor terms, respectively".

P4 15 Add an equation, explanation or reference how B_AT is calculated.

P4 L27-28 SIA and SSA are not yet defined.

P5 L5 "horizontal derivatives" and "vertical derivatives" of what? Clarify.

P6 L16 How is the water pressure defined? Add equation or reference.

P6 L25 So "beta = C*N" for temperate ice. What is assumed for the rest? Clarify.

P6 L25 I expected to get some information on how C_f is calibrated in this paragraph, maybe just a list of options that could be used, then you get back to that later.

P6 L29 Avoid use of "flux correction" as it has a specific meaning in coupling climate models. Use e.g. "flux calculation" as on top of the next page.

P7 L18-19 Here it would be good to already know how the grid is laid out. Where is the velocity defined compared to the ice thickness nodes. You later state that you are using a "staggered Arakawa C-grid", but it may not be clear to all what that implies. Could you add a clear description or figure where the different quantities are defined? This would also help in the following to explain the numerical implementation of certain schemes.

P7 L19. So the flux is imposed on two grid points in both x and y direction and applied there simultaneously? More precision is needed to make clear how to implement this.

P7 L22 Wouldn't this give back force at the places where velocities are calculated, not at the GL where it is needed?

P7 L27 Replace "Ice front" by "Iceberg" in front of "calving".

P8 L1 The section header states "Temperature coupling". There is no description of coupling in this section, only how the temperature is calculated. Maybe change title to "Thermodynamics" or "Ice temperature calculation".

P8 L2 Could give a sentence of introduction to state that this is the classic way to solve thermodynamics and similar to many other models (references). Or is there anything special here that I have overlooked?

P8 L2- Consider to give some indication on how all of this is solved numerically. How are the differential equations discretised? Which numerical schemes are used for advection and diffusion (upwind, second order, Lax)?

P10 L12 Hardly any information is shared on how the given equations are solved numerically. I believe it would make the paper a much more interesting reference for other modellers and even people in your own group if some details would be added on the practical side of the modelling.

P10 L13 More precision is needed to understand what variables are defined where on which (staggered) grid, also in the vertical. How is the vertical grid laid out? Is the order up-down or down-up? Is the first vertical grid point from the top where T is solved assumed at the boundary or representing the middle of a first layer? How is that at the base?

P10 L17 "the resolution is *reduced* to ...".

P10 L19 Replace "computes" by "uses" or "computes with". Add "which is dynamically calculated" after "(Eq.2)" or similar.

P10 L24-29 Could this be visualised for clarity?

P10 L28 How small is "small" in this context? Clarify

P11 L1 Add a reference for OpenMP.

P11 L15 Add a short overview what is coming next before going into details.

[Figure]

P11 L20 Add reference for ISMIP6 initMIP-Antarctica (ISMIP6 paper, website).

P12 L6 Is K0 changing the basal drag, or the basal drag coefficient? Clarify.

P12 L10 How does the BMB field keep the ice shelves stable? Clarify.

P12 L15 In my mind, the basic idea of LHS is to *sample* the hypercube and not perform all possible experiments. Maybe reword to avoid "the whole cube" if this is correct.

P12 L19 How is the low resolution data set produced from the original Bedmap2 data? Direct subsampling or smooth interpolation? This is a crucial part of preparing the input data and should be treated with detail and precision.

P12 L20 Reword "discarded from the ensemble" to "not explored in this ensemble"

P13 L12 If differences are below 500, why does the scale go to 1000 in the figures?

P13 L18 Do all these models use the same data, the same processing to get to the final input data and have similar resolution? Do you know if this is an error in the data or a problem of coarse resolution? What is different in models that do not show these features, if there are any?

P13 L20 If the last point is resolved, maybe "... suggesting a common source of error related to the coarse model resolution". Or similar to add some interpretation to this comparison.

P16 L3 How can you be sure that the parameter range is sufficient/optimal for the transient experiments? Please add a short discussion on that.

P16 L10 Maybe "post-LGM retreat"?

P17 L4 "relative to observations ... in this case, where ..." and briefly remind us what is different in the present experiments.

P17 L5 Where is the northern part of East Antarctica?  polewards = south!,
north=towards the margin?

P17 L15 Inversion of what? Specify.

P17 L23 This comes a bit unexpected. Has vertical temperature been worked on in this paper? Why only the vertical?

P17 L33 "lead to a more dynamic grounding line position"

P17 L34 Has sensitivity to SL changes really been discussed in the paper?

P17 L34 Add sensitivity to sub-shelf melt rates?

P18 L11 Basal hydrology and semi-lagrangian tracking are described in "2.2 Additional features" next to other aspects that are not mentioned here in the conclusions. It may be useful to make it clearer already in the main text which of the described features are new developments in GRISLI.

P18 L26 Why does validating a model for the Antarctic give confidence to also use it for the NH. A bit more information is needed here to bridge that gap.

P18 L30 Replace "and" by "or", unless all three have to be contacted to get the model code.

Tables and figures

P26 in the middle. Replace "N.m" by "N m"?

P27 caption Table 2. Write out LHS.

P27 Figure 2. Left panel cold have additional contours to delineate the regions and a colour bar. Why does the grounding line have different melting rates than the shelf?

P29 Figure 5. Use a colour for the GL contour that does not appear in the colour map, e.g. black or dark gray.

P30 Figure 6. Same as for figure 5.

P32 Figure 8. Suggest to move the parameter values to a table or the main text.

P34 Figure 12. "materialised" –> "shown" or "indicated" Why is the GL so patchy here compared to the steady state case?

P35 Figure 13. Same as for figure 5.

―――――――――――――――

---

## Author Comment (AC1) · 7 Sep 2018

*Overall: This paper provides a description of the current GRISL1 ice sheet model, focusing on modifications and extensions from the previously described version in 2001. The model is designed for relatively coarse-resolution long-term paleo applications. Main change from 2001 include the specification of grounding-line fluxes (Schoof, 2007), and a basal hydrology model. A large ensemble with Latin Hypercube sampling is used to constrain and calibrate four uncertain model parameters. The paper is clear and provides useful supporting documentation for GRISL1 users and background for other papers on GRISL1 applications.*

Thank you for your time in reviewing our work. In the following we provide a point by point response to your comments. Referee comments are italicised and in orange.

*Specific comments:*

*1. There is considerable scatter in Fig. 7, showing pair-wise parameter correlations with the results (here, rms error in modern ice thickness). This may be because the Latin Hypercube (LHC) sampling of the large ensemble (LE) may be too coarse to meaningfully detect pair-wise dependencies. The quasi-random distribution of red, blue and green stars in these panels is reminiscent of corresponding figures in Applegate et al. (2012, The Cryo., their Fig. 1). Chang et al. (GMD, 2014) subsequently found that the scatter in the Applegate study is due to inadequate sampling in the high-dimensional parameter space, and they used additional statistical analysis with Gaussian emulation to extract meaningful dependencies (their Fig. 4a vs. 4b). That study had a similar number of parameters (5) and ensemble members (100) as here (4 and 150). In a similarly sized Antarctic LE, Pollard et al. (GMDD, 2016) found that meaningful dependencies could only be found with "full-factorial" sampling, i.e., a run for every possible combination of parameter values, requiring 5ˆ4 = 625 runs for 4 parameters and 5 values each. If that many runs could be performed here, it might yield much more meaningful pair-wise results than in Fig. 7. However, if that would be too computationally expensive, it could be left to future work, and the above caveats could just be noted.*

Encouraged by your concern we doubled the size of the ensemble. We now have 300 members for each formulation of the flux at the grounding line (600 members in total). Our ensemble is now considerably larger than the one of Applegate et al. (2012) considering that we have 4 parameters instead of 5. In addition, we have added the RMSE information in some figures (former Fig. 4 and Fig. 7) in order to facilitate the emergence of relationships. With this larger ensemble, the RMSE in the parametric space (new Fig. 7 and Fig. 8) mostly confirms what was suggested in the first version of the manuscript: there is generally no clear relationship between our parameters that can explain the RMSE with the major exception of the variable E_SIA versus C_f for simulations using Schoof et al. (2007) and E_SIA only for simulations using Tsai et al. (2015). From this, we suspect that to obtain meaningful dependencies we should in fact probably expand drastically the ensemble, probably by doubling again its size (order of magnitude from $10^2$ to $10^3$). This is beyond the scope of the manuscript but we acknowledge the fact that emulators trained with very large ensemble are a very promising field of applications for large scale ice sheet model.

*2. The simulations here use uniformly prescribed basal drag coefficients, and do not use an inversion method to deduce a spatial map. There is discussion on the pros and cons (pg. 6, 15, 17), which makes good points for not using inverse methods. But it does not mention the primary motivation (I think) for using them: that without them, modern errors in ice thickness are much larger (as in Figs. 5,6), and can be made much smaller using an inverse procedure. Since these*

*errors are the primary metric here for evaluating the model, this could help to make the calibration of model parameters more meaningful. I think the whole issue depends on whether the inverse-produced map captures real bed variations at all, or if it just cancels with and obscures other physical errors in the model. I suggest mentioning this within the existing discussion. Also, the point made on pg. 17, line 16, on the desirability of making basal coefficients a property solely of internal model parameters (such as N here, in Eq. 14), is debatable: apart from basal temperatures and water amount of course, spatial variations in basal sliding can also depend importantly on geologic bed type, roughness, and the distribution of deformable till, which are outside the scope of the model.*

The simulations in this paper do not use uniformly prescribed basal drag coefficients as they are computed from the effective water pressure that varies both in space and time (Eq. 16). This is precisely the capability of this formulation to respond interactively to changes in geometry that motivated our choice over an inverse method.

Using an inverse method to infer the basal drag coefficients would not help in calibrating the model parameters because, by construction, the inverse method will correct any bias in both the forcings (climate/bedrock) and the model physics. For example, for different enhancement factor values we can, in principle, infer different basal drag coefficients maps that will result in close-to-observation geometry.

We have nonetheless added a sentence on the pros of inverse methods (Sec. 2.1.3):
"Inverse methods are particularly adapted to produce an ice sheet state (e.g. geometry and/or velocity) close to observations. However, such methods do not provide [...]"

About a basal drag computed from internal variables only, you are right mentioning the importance of bed properties such as geologic bed type and roughness. These are largely uncertain both in term of their values and in term of their impact on ice dynamics. We have slightly modified these lines:

"A step forward would be to use the basal drag computed from inversion in order to deduce a formulation based solely on internal parameters. Amongst these parameters, along with the basal effective pressure, the large scale bedrock curvature and/or sub-grid roughness could be used, similarly to Briggs et al. (2013). However, some key basal features, such as the geologic bed type and the deformable till distribution, remain today largely unknown below present-day ice sheets and will contribute to large uncertainties in the basal drag formulation."

*3. In Eq. 14 on pg. 6, and section 2.2.1 (Eqs. 24-26, pg. 9), it is not clear how some variables for basal hydrology are determined: h_w or p_w (which are related, line 24), and effective pressure N needed for Eq. 14. Presumably there is a prognostic equation in the hydrology model for h_w, i.e., d(h_w)/dt = ..., that is not shown here. Perhaps it is the equation mentioned on pg. 9, line 23. Also N possibly depends on p_w. This information, and the equation for h_w, should be included. (Incidentally, if N depends on depth below sea level as in several other models, I would question how can it reasonably depend on that, at distances 10's or 100's km inland from the grounding line).*

We acknowledge that the original description of the hydrology was somehow incomplete and we have considerably rewritten this section with clarity in mind. The prognostic equation for the hydraulic head h_w is now presented. We also explicitly mention how we compute p_w and N from h_w. N does not depend explicitly on the depth below sea level as it is computed as the difference from the ice load pressure (rho_i g H) and the basal water pressure (rho_w g h_w). However, the

depth below sea level is a necessary boundary condition for the hydraulic head at the marine ice sheet margin (h_w=sealevel-Bbed).

*4. The determination of buttressing factor phi_bf in Eq. 15 (pg. 7) is an important part of the use of the Schoof flux equation, but the procedure is unclear to me from lines 21- 25 on that page. Perhaps the first solution provides the back-stress-free solution...does that solution use Eq. 15 with phi_bf = 1? Then what is the second solution, and where does its value of phi_bf come from? These questions may not make sense, and just show my confusion. Hopefully this paragraph can be clarified, and perhaps expanded if that would help.*

We acknowledge that it was not clear in the first version of the manuscript. In fact, in our framework we compute the velocity equation three times with the two first iterations being used to compute phi_bf. The first iteration is computed on the simulated geometry with no flux adjustment at the grounding line. The second iteration is computed the same way, except for the fact that the ice shelves are assigned to a very low viscosity so that they cannot exert any back force. The buttressing ratio phi_bf is then computed as the velocity ratio between these two computed velocities. The flux adjustment at the grounding line is only applied for a third iteration which gives us the actual velocity field. We have made this clearer in the manuscript:

"To evaluate the back force coefficient phi_bf, we solve the velocity equation twice. The first iteration is computed on the simulated geometry with no flux adjustment at the grounding line (i.e. not using Eq. 17 nor Eq. 18). The second iteration is computed the same way, except for the fact that the ice shelves are assigned to a very low viscosity so that they cannot exert any back force. The buttressing ratio phi_bf is then computed as the velocity ratio between these two computed velocities. Once phi_bf is estimated, we solve the velocity equation again, this time accounting for the flux ajustment at the grounding line using Eq. 17 or Eq. 18, in order to estimate the velocity used in the mass conservation for this time step."

We acknowledge for the fact that the two first iterations produce unrealistic simulated velocities as they do not account for any specific treatment at the grounding line. However, we assume that the ratio in velocities is representative for the buttressing effect of ice shelves.

*5. pg. 16, lines 5 to 7: Perhaps, the timings of the deglacial retreat in AN40T vs. AN40S can be assessed vs. papers in the RAISED reconstruction volume (Bentley et al., 2014), or other data, in order to determine which one is more realistic. The paper seems to decide rather arbitrarily that the AN40T case is more realistic (pg. 18, line 25).*

The fact that the model is still drifting at +10 kyr in the future with AN40S is a clear indication of a too slow retreat in this case. However, it is true that our climatic forcing is relatively simple and that with an alternative climate forcing we could maybe have a faster retreat with AN40S. Keeping that in mind, we have moderated this sentence:
"This suggests that, in our model and under the climate forcing scenario we use, the Tsai et al. (2015) formulation produces a more realistic grounding line retreat rate."

There is unfortunately no archive that allows for an ice volume change reconstruction of the Antarctic ice sheet during the last deglaciation. While, the RAISED reconstructions do not quantify the change in ice volume, it is indeed nonetheless, at present, the most complete data compilation on the extent of the grounding line during the last deglaciation. However, the temporal resolution (5 kyr) together with the fact that the largest uncertainties remain in the Weddell and Ross seas, make

it difficult to compare with our model results. This is why although we discuss the RAISED reconstructions in the original version of the manuscript for the ice extent at the last glacial maximum, we did not use this as a constraint for the timing of the deglaciation.

*6. pg. 13, lines 9-10: The pairs of values "1.5 to 3" and "1.5 to 5" do not seem to relate to the bottom-right panels in Figs. 3 and 4, for basal-drag coefficients K0 (which are being discussed in that sentence). They seem to relate better (but still fuzzily) to the bottom-left panels for enhancement factors E_sia.*

The initial formulation was misleading as we were indeed referring to the enhancement factors. We reformulated as:
"As a consequence, for the AN40T ensemble, the enhancement factor requires values between 1.5 and 3 in order to reach a good agreement with observed ice thickness, whilst values within 1.5 to 4 are acceptable for AN40S."

Technical comments:

*The English usage is generally good, but isolated words or phrases could be improved/corrected, some of which are noted below.*

*pg. 1, line 19: Change "are evidences" to "is evidence".*

Done.

*pg. 1, line 23: Change "An other" to "Another".*

Done.

*pg. 2, line 7: I think "prograde" should still be "retrograde", for MICI as well as for MISI.*

We acknowledge the fact that a retrograde slope will inevitably amplify both the MISI and MICI. However, as postulated by Pollard et al. (2015) and contrary to the MISI, the MICI can also occur on neutral and prograde slopes. We clarified this sentence:
"Additional instabilities may also occur on neutral/prograde bed slopes in relation with the structural instabilities of tall ice cliffs (marine ice cliff instability, MICI, Pollard et al., 2015)."

*pg. 2, line 23: The word "diffusion" should probably be removed (?).*

Removed.

*pg. 3, line 14, and several later places: "Tab. 1" should perhaps be "Table 1".*

The GMD manuscript preparation guidelines for authors suggest to use abbreviations (Sec., Fig., Eq., Tab.) when used in running text unless it comes at the beginning of a sentence. We have followed these guidelines consistently.

*pg. 4, line 1: The use of two "respectively"'s in the same sentence is confusing - perhaps divide into 2 separate statements for sigma_i and tau_ij.*

It has been changed to:
"[…] where tau_ij=x,y,z are the shearing stress tensor terms and sigma_i=x,y,z the longitudinal stress tensor terms, defined as (i=x,y,z):
sigma_i = tau_ii"

*pg. 4, line 26: Change "Alike" to "Like".*

Done.

*pg. 4, lines 27-29: The word "reduces" in line 27 seems to contradict the word "favour" in line 29. Or perhaps "longitudinal" should be "shearing" in line 29 (?).*

Thanks for noticing, it was effectively a mistake. This part has been expanded and reformulated.

*pg. 5, line 24: What does "see also numerical feature" mean?*

Changed in the text to:
"[…] see also Sec. 2.3 on the numerical features"

*pg. 16: I would suggest emphasizing, as a positive note, that if the (interpolated) grounding line position is known, then all that is required to obtain ice thickness H_gl at the grounding line for Eq. 15 is (1) bedrock bathymetry interpolated to the grounding line position, and (2) sea level. (This is because of the floatation criterion at the grounding line of course).*

The sub-grid position of the grounding is known because we linearly interpolate the floatation criteria based on the knowledge of thickness, bathymetry and sea level on the centred GRISLI grid. For more clarity for the reader, we have added a schematic representation of the staggered grids (Fig. 2) and expanded the description of how we apply the grounding line flux on the velocity nodes.

*pg. 10, line 23-24: Explain the need for the artificial extension (to get an ice front parallel to x or y).*

With this extension, the front of the ice shelf is always parallel to either x or y which facilitates the application of boundary conditions. This is now explicitly stated in the manuscript. A schematic representation of the different cases for the elliptic equation is now shown in Fig. S1 of the supplementary material.

*pg. 11, line 20: Misspelled "projet".*

Corrected.

*pg. 12, line 15: "in 150" should be "of 150".*

Changed.

*pg. 12, line 20: Perhaps change "are discarded from" to "are not included in" ?*

It has been changed to "are not explored in".

*pg. 14, line 6: Change "somehow" to "somewhat".*

Done.

*pg. 15, line 16: Does this mean that a 100-kyr long spinup is performed with perpetual modern climate for every ensemble member? If so, say that more clearly.*

All the ensemble members (300x2) in Sec. 3 are 100-kyr integrations of the model using perpetual modern climate (p.11 l.16-17 in the initial manuscript). The spun-up ice sheet used for the transient simulations is the final state obtained at the end of the 100-kyr integration in Sec. 3. For sake of clarity, we reformulate as:
"We used the 100-kyr integration under perpetual modern climate in Sec. 3 as a spin-up for the transient simulations."

*pg. 15, line 24: The range of 10 to 20 m eustatic sea level drop here is actually a bit larger than several recent model studies. This might be due to larger basal drag coefficients used here on modern continental shelves, so when grounded ice expands onto them at LGM, the expanded ice is thicker there.*

The reviewer is perfectly right here. During glacial periods, the ice sheet expands onto part of the continental shelf that presents presumably different bedrock conditions. In particular, we can expect to find more deformable till relative to hard bed in these areas, facilitating the ice flow for large part of today ice-free regions. For these reasons, some authors choose to use a two-value basal sliding coefficient for hard bedrock (bedrock above sea level) and deformable sediments (bedrock below sea level) (e.g. Pollard and Deconto, 2012). Because geologic information below present-day Antarctica is poor, we have preferred to keep our approach as simple as possible with no additional tuning. However, for future work it is clear that sensitivity studies on the role of bedrock characteristics will have to be performed.

Following your comment, we have added the following in the manuscript:
"Our reconstructions are nonetheless at the higher hand of recent studies. This could be related to the fact that we do not account for different geologic bed types between today ice-free (with extensive amount of deformable till) and glaciated (mostly hard bed) continental shelf. To account for this, some authors have chosen a two-value basal drag for these different regions (e.g. Pollard and Deconto, 2012). Because of the large uncertainties related to the bed properties we have decided to ignore these differences, keeping in mind that this can bias our results towards thicker ice sheet when the ice expands over the continental shelf."

*pg. 16, line 4: Change "In turns" to "In turn".*

Done.

*pg. 17, line 2: "using an inverse method" sounds like one is used here. Make it clear that one is not, and that phrase refers just to the references earlier in the sentence.*

This now reads:
"We have presented results from the updated version of the GRISLI model. Whilst the model is able to reproduce present-day Greenland (Le clec'h et al., 2017) and Antarctic (Ritz et al., 2015) ice sheets when using an inverse method to estimate the basal drag, our simulations with an interactive

basal drag computed from the effective pressure show some important disagreements relative to observations."

*Table 1: Some of these variables do not seem to be used in the text, e.g., those under "Deformation". Others may have a different name, e.g., h_till.*

We have now given more information on the computation of the viscosity in the model and we refer explicitly to the variables listed below "deformation". We have checked the table carefully and corrected a few mistakes. If some of the variables here are effectively not used in the text, in particular the ones for the isostatic rebound model, we have nonetheless preferred to keep them here for documentation of this particular version of the model.

*Fig. 2: The relationship between the sector boundaries (left-hand panel) and the contour divisions for basal melt rates (right-hand panel) is confusing, not as one might expect. That is, there seems to be some divisions between the colors in the right-hand panel that are not present in the left-hand panel, and vice-versa.*

We imposed a specific (high) sub-shelf melting rate for the deep ocean (depth greater than 2500 m). This is why one region (deep ocean) of the right-hand panel does not appear on the left-hand panel. We now explain this in the text:
"Sub-shelf melting rate for the deep ocean (depth greater than 2500 m) are assigned a value of 5 m/yr."
Also, if some sectors that appear on the left-hand panel do not appear on the right-hand panel this is because they have the same or similar sub-shelf melting rates and cannot be distinguished.

**References**

Pollard, D. and DeConto, R. M.: Description of a hybrid ice sheet-shelf model, and application to Antarctica, Geosci. Model Dev., 5, 1273-1295, https://doi.org/10.5194/gmd-5-1273-2012, 2012.

Pollard, D. and DeConto, R. M. and Alley, R. B.: Potential Antarctic Ice Sheet retreat driven by hydrofracturing and ice cliff failure, Earth and Planetary Science Letters, 412, 112-121, https://doi.org/10.1016/j.epsl.2014.12.035, 2015.

Bentley, M. J., Ó Cofaigh, C., Anderson, J. B., Conway, H., Davies, B., Graham, A. G. C., Hillenbrand, C.-D., Hodgson, D. A., Jamieson, S. S. R., Larter, R. D., Mackintosh, A., Smith, J. A., Verleyen, E., Ackert, R. P., Bart, P. J., Berg, S., Brunstein, D., Canals, M., Colhoun, E. A., Crosta, X., Dickens, W. A., Domack, E., Dowdeswell, J. A., Dunbar, R., Ehrmann, W., Evans, J., Favier, V., Fink, D., Fogwill, C. J., Glasser, N. F., Gohl, K., Golledge, N. R., Goodwin, I., Gore, D. B., Greenwood, S. L., Hall, B. L., Hall, K., Hedding, D. W., Hein, A. S., Hocking, E. P., Jakobsson, M., Johnson, J. S., Jomelli, V., Jones, R. S., Klages, J. P., Kristoffersen, Y., Kuhn, G., Leventer, A., Licht, K., Lilly, K., Lindow, J., Livingstone, S. J., Massé, G., McGlone, M. S., McKay, R. M., Melles, M., Miura, H., Mulvaney, R., Nel, W., Nitsche, F. O., O'Brien, P. E., Post, A. L., Roberts, S. J., Saunders, K. M., Selkirk, P. M., Simms, A. R., Spiegel, C., Stolldorf, T. D., Sugden, D. E., van der Putten, N., van Ommen, T., Verfaillie, D., Vyverman, W., Wagner, B., White, D. A., Witus, A. E., and Zwartz, D.: A community-based geological reconstruction of Antarctic Ice Sheet deglaciation since the Last Glacial Maximum, Quaternary Science Reviews, 100, 1-9, https://doi:10.1016/j.quascirev.2014.06.025, 2014.

---

## Author Comment (AC2) · 7 Sep 2018

**F. SAITO (Referee #2)**

*This paper describes the numerical ice-sheet model GRISLI ver 2.0, in particular, with the application for the Antarctic ice sheet simulation. I think this paper is fairly well written with some exception below, and can be accepted with minor revision.*

*One thing better to include is technical (numerical) procedures and properties adopted in the model. Since the source code is not opened to public, information how to solve the model equation is useful for those who has not contact with the model. For example, how to solve the Eq.(2), (thickness evolution)? Explicit, implicit or others? How to solve the linear equations, direct, alternate-direction, conjugate gradient, or other method? How to solve the non-linear ice-shelf equation (Eq.12)? Linearize them? Or the velocity-dependent viscosity (\bar{eta}) at the previous time step is used? How to determine the convergence of the solutions where the iterative solver is involved? Such details are all necessary to evaluate the numerical accuracy of the model, if they want. I suppose they are more or less the same as the original version of the model (Ritz et al. 1997,2001), but repetition (or at least citation of the old papers) are needed for completeness of the model description.*

This is perfectly in line with Referee #3 general comment and we acknowledge that the initial version of the paper provided only few technical information. In the revised version, we now explicitly show the staggered grid in Fig. 2. We have also added two additional figures in the supplementary material showing: i) the boundary conditions when extending the front to the edges of the geographical domain and ii) the matrix used to solve the elliptic equation. We have also considerably expanded Sec. 2.3, adding more information on the numerical resolution of the equations:

"The mass balance equation is solved as an advection-only equation with an upwind scheme in space and a semi-implicit scheme in time (velocities at the previous time step are used). The numerical resolution is performed with a point-relaxation method with a variable time step. The value of this time step is chosen to ensure that the matrix becomes strongly diagonal dominant to achieve convergence of the point-relaxation method. The criteria is thus a threshold that is inversely proportional to the fastest velocity on the whole grid. Note that this smaller time step is solely used for the mass conservation equation and subsequent variables (e.g. surface slopes, SIA velocity) while the rest of the model uses a main time step, typically ranging from 0.5 to 5 years depending on the horizontal resolution.

To solve it, the ice shelves/ice streams equation (Eq. 14) is linearised. The viscosity is computed using an iterative method starting from the viscosity calculated from strain rates from the previous time step. As this equation is the most expensive part of the model, the iteration mode is not always used depending on the type of experiment (for instance not crucial when the objective is to reach the steady state). In this case the viscosity of the previous time step is used. The linear system is solved with a direct method (Gaussian elimination, sgbsv in the Lapack library (www.netlib.org/lapack)).

For the temperature equation (Eq. 19), we solved a 1D advection-diffusion equation for each model grid point. The resolution is performed with an upwind semi-implicit scheme (vertical velocity and heat production at the previous time step is used). The ice thermal conductivity is computed as the geometric mean of the two neighbouring conductivities (Patankar, 1980). Because the horizontal diffusion is neglected, the only horizontal terms concern horizontal advection and are computed

with an upwind explicit scheme. The heat production is computed at the velocity (staggered) grid points and is then summed up to the temperature (centred) grid points."

*Another thing better to rewrite is the lateral boundary condition of the ice shelf, which is still unclear to me. As described in Sec 2.3, the ice shelf is extended towards the edges of the model domain. As far as I understand, to remove row i from the matrix corresponds to set horizontal velocity as zero at the front.*

No, it does not mean that horizontal velocity at the front is zero, it means that it has an undefined value.

*I am not sure this is what the authors expect. I suggest to rewrite the second paragraph of the Sec.2.3 to clarify how to formulate the matrix in the model.*

We now show the matrix in the supplementary material and we have been more specific in the revised version, which now reads:

"The resolution of the elliptic system (Eq. 14) is the most expensive part of the model. This is further amplified by the way we prescribe boundary conditions. As in Ritz et al. (2001), the ice shelf region is artificially extended towards the edges of the geographical domain. This artificial extension does not have any consequence on ice shelf velocity since added grid points (that we call "ghost" nodes) are prescribed with a negligible ice viscosity (1500 P a s). The front is then parallel to either x or y (Ritz et al. 2001) and thus the boundary condition there is easy to implement (see also Fig. S1 in the supplement). The boundary condition at the real ice shelf front is implicitly done by solving (Eq. 14). However this method increases substantially the size of the linear system solved in (Eq. 14). To go around, a simple reduction method is implemented. Eq. 14 can be written as $\tilde{A}\ \tilde{u} = B$ where $\tilde{u}$ is a vector alternating $u_x$ and $u_y$ components for all the velocity grid points, $\tilde{A}$ is a band matrix (very sparse) and $B$ is a vector corresponding to the right hand terms in Eq. 14. Every line of $\tilde{A}$ and $B$ are scaled so that the diagonal terms of $\tilde{A}$ are equal to 1. If, for a given velocity node, all the non diagonal terms of the column are very small compared to 1, this means that this node is actually not used by any other velocity node and this line of the matrix can be removed. The threshold to neglect nodes is related to the value of the integrated viscosity of "ghost" nodes . In practice, given its size, the matrix $\tilde{A}$ is not actually fully constructed, only the non zero sub/sur diagonals are. An illustration of the matrix is shown in Fig. S2 in the supplement. "

*In addition, I definitely agree to the specific comments 3 and 4 by the referee #1. The authors should clarify the formulation and the procedures to compute basal hydrology and the back-force coefficient.*

The basal hydrology is now presented in much more details, including the prognostic equation for the hydraulic head, h_w, from which the effective pressure is computed. We have also clarify in the revised version how the back-force is calculated (please see also our response to the 4th point of referee #1).

*Some minor points (PmLn corresponds to the line number n in page m)*

*Units: use \unit{} macro.*

Thanks for the suggestion, we have done so.

*P1L6: 'or Tsai et al....' may be better?*

Changed.

*P2L16: 'right' might not be a right word for this context. How about 'practical tool for....'*

This has been replaced by "the most suitable tool for".

*P3L19 Eq(1): Divergence, not Gradient (need dot).*

We have followed this convention consistently in the text.

*P3L22 Eq(2) and after: \bar{u}_x is better than \bar{u_x}.*

Changed.

*P3L26 Eq(3) and similar array equations: Use \displaystyle.*

We have followed your suggestion.

*P4L1 or around: Need definition like \sigma_i = \tau_{ii}, otherwise the paper misses the equation for longitudinal stress components, since Eqs 5, 6, 8, 9 are described with \tau_{ij}.*

We agree, this has been added.

*P4L11 Eq(7) or (9): The enhancement factor should be inserted. Otherwise E_{SIA} in P12L1 is confusing.*

We have now added the equation for B_{AT} in which the enhancement factor appears (renamed as Sf for consistency with Ritz et al., 1997 and to avoid confusion with the activation energy Ea).

*P4L16 Eq(8): Need range of i,j.*

Information added (i,j={x,y,z}).

*P4L25 Eq(9): No explicit formulation of B_{AT1} and B_{AT3}. Are they documented in Dumas (2002)?*

We have added the equation for B_{AT} and expanded the description of how they are calculated for the Glen viscosity and the linear viscosity. We hope that the new version of the manuscript contains all the necessary information.

*P5L10 Eq(10): i in \rho_{i} conflicts with row i.*

The density of ice is simply referred as rho consistently in the whole manuscript.

*P5L9: (for i = x,y) not (for i=x,y,z). When i=z in Eq(11), the coefficient 2 must disappear.*

Thanks for noticing, this has been corrected now. Vertical velocity is mentioned later in the manuscript.

*P5L10 Eq(11): no definition of B. B may conflict with B_{AT}, which is better to avoid.*

We have added the definition of B (bedrock elevation). We stick to B and B_{AT} for consistency with Ritz et al. (2001).

*P5L21: S is already defined in L9.*

Thanks for noticing, the definition here has been removed.

*P5L23: (see also 2.3 numerical feature)*

Changed in the text to:
"[…] see also Sec. 2.3 on the numerical features"

*P5L26: The basal drag is very small but not necessarily zero. All we can do is to neglect it.*

We agree, we now simply say that we neglect it.

*P7L12 Eq(16): No definition of H_g. Typo of H_{gl}?*

Thanks for noticing, we meant H_{gl}.

*P8 Sec2.1.6. Need to mention how to compute the vertical velocity. I suppose vertical velocity is not directly computed as Ritz et al. (1997).*

Vertical velocity is indeed computed as in Ritz et al. (1997). We added this information in the text :
"$u_z$ is the vertical velocity, computed as in Ritz et al. (1997) (wt in Eq. 14)."
This velocity is used for 3D advection (temperature and tracers).

We have also added more information on the link between temperature and viscosity:
"The viscosity for the velocity grid points is the horizontal average of the viscosity on the centred grid and not the viscosity computed from the horizontal average of the temperature. This is preferable for regions with mixed frozen and temperate basal conditions."

*P8L10: 'zero' requires the unit. I prefer to write as 'the melting point'.*

True, it has been changed.

*P8L25 Eq(22) Write \exp (backslash before exp) following LaTeX convention.*

Done.

*P9L27, Better to cite Le Meur and Huybrechts after ELRA sentence also.*

Done.

*P10L26: Better to avoid to use A and B for matrix and vector, which conflict with the rate factor or bedrock elevation.*

Most of the letters in the alphabet relate to a variable used at some point in the model. We think that A and B, being the first letters, are more generic. We have nonetheless added a tilde (~) on top of them for a better distinction with respect to the rate factor and the bedrock elevation.

*Table 1: Unit of the acceleration should be m/s. Really same values for ice and mantle thermal conductivity?*

Thanks for noticing this. We changed the unit of the acceleration and we corrected the ice thermal conductivity.

*Figure 1: If z at the ice bottom is always zero as the figure, you need to reformulate all the governing equations using z.*

Sorry for this, z should have been B at the ice bottom and S=B+H at the top. The figure has been updated.

---

## Author Comment (AC3) · 7 Sep 2018

*Summary: This paper describes a new version of the GRISLI ice sheet model, including new model development since an earlier version many years ago (Ritz et al., 2001). While there are no major scientific advances relative to the state-of-the-art manifested with this update, documenting the current state of the model and the individual progress that has been made is well appreciated. The paper is generally well written and clear. Nevertheless, I believe there could be more detail given in some of the descriptions to make it a better reference and make the paper more accessible for other modellers. I consequently recommend publication in GMD with some corrections as detailed below.*

Thanks for your encouraging comment. We provide a point by point answer to your concern in the following.

*General comments:*

*I believe it is a good practice to (regularly) publish model description papers like the present one, to document the applied models, increase transparency and allow for other modellers to learn, improve upon and critically evaluate the applied techniques. One point I find regrettable with the present paper is that the authors seem to not plan to publish the model code alongside with the manuscript. I know that this may not be common practice in our community, but I believe it would be an important step forward. I would applaud if the authors would think about how to make the code publicly available, possibly with certain restrictions.*

We agree with the reviewer on this, however in practice it is not that trivial to make GRISLI code published publicly. The public distribution needs to be done with a specific license (e.g. GNU public licence). Such a license is currently lacking in GRISLI. The model has been developed from the 90ies and has benefited from the additions from numerous contributors. To put a license to the current model, we have to get the permissions to all the contributors from the past 20 years. To date, we do not have this permission from all the contributors. However, as stated in the manuscript, any potential users are encouraged to get in touch with C. Dumas, A. Quiquet or C. Ritz, to start a collaboration.

*While the applied modelling techniques are mostly well described in words and equations (textbook style), the numerical implementation is often not possible to determine. I invite the authors to make an additional effort to increase the precision. This is even more important when the model code cannot be consulted. The ultimate goal should be that someone who does not know the model would be able to implement a specific feature from the given information. See also specific points below.*

In the revised version of the manuscript, we have expanded on the different subjects: 1) the description of the polynomial flow law and the link between temperature and viscosity; 2) the computation of the flux at the grounding line; 3) the structure of the grids. In addition, we have given more information on the numerical resolution of the equations in the "numerical features" section (Sec. 2.3).

*Reference to textbooks and earlier works is mostly in order. However, in some cases it would be useful for the reader to have some additional ('meta') information for the specific descriptions. E.g. who else is using the same technique, or does the applied technique represent a notable difference/novelty compared to other models used in the community. If other approaches exist, do you have reasons to choose this approach compared to another and why (simplicity, better results,*

*tried other approach but didn't work ...). Again, the motivation should be to make the paper a useful reference and interesting resource for another modeller trying to implement a specific feature.*

Because the novelties were not presented in a sufficiently clear way in the original manuscript, in the revised version, we have systematically referred to the similarities and differences from the previous model description (Ritz et al., 2001). We also have referred to the similarities to other models (SSA velocity as a sliding law, temperature computation, particle tracking scheme, etc.).

*Specific comments:*

*P1 L14 Which time scales or time scale range, be more specific.*

From diurnal to multi-millenial. Added in the text.

*P1 L15 "surface albedo" is not a feedback, nor "freshwater flux". Please clarify.*

Changed for: "[...] through multiple feedbacks such as temperature - surface albedo, gravity waves and oceanic circulation changes related to freshwater flux release."

*P1 L18 Not sure about the connection between the two sentences implied here.*

Changed for:
"If the two major ice sheets have been mostly stable for at least the last 1000 years, their contribution to global sea level rise in the future is largely uncertain. Conversely, in the past, there is evidence of sea level rise as fast as four metres per century (e.g. Fairbanks, 1989; Hanebuth et al., 2000; Deschamps et al., 2012)."

*P1 L21 I would suggest rewording to avoid drastic terms "rapid" and "destabilisation".*

We agree with the reviewer that such terms are often used deliberately in inappropriate context. However, our previous sentence directly refers to the melt water pulse 1A event, which leads to abrupt sea level rises as high as 4 m within a century. This event can certainly be qualified as "rapid" and is induced by an ice sheet destabilisation.

*P1 L22 "The surface mass balance-height feedback has ..."*

Thanks, we have followed your suggestion.

*P2 L1-3 Not all bedrock in the Antarctic shows a retrograde slope. More precision needed.*

We have added "[…] is related to the fact that large parts of the bedrock presents a retrograde slope [...]"

*P2 L4 MISI driven retreat does not have to be very fast. Suggest removing "fast and"*

Done.

*P2 L7 Add "ice" before "cliff"*

Done.

*P2 L7 Buttressing already decreases when the ice shelve is removed and a reason for ice cliffs to fail. Reformulate.*

True, we have reformulate as follow: "Additional instabilities may also occur on neutral/prograde bed slopes in relation with the structural instabilities of tall ice cliffs (marine ice cliff instability, MICI, Pollard et al., 2015)"

*P2 L10 What is the range of temporal and spatial scales, specify.*

Changed to:
"Because they include processes operating on variety in temporal, from diurnal to multi-millenial, and spatial scales, from a few metres to thousand of kilometres, [...]"

*P2 l10 "temporal and spatial scales" of what exactly?*

Now specified, please see our response to your previous comment.

*P2 L14 There are also state of the art models that are not FS, clarify.*

In this sentence, the term "state of the art" has been replaced by "the most comprehensive".

*P2 L19 Remove "Conversely".*

Done.

*P2 L21 add "e.g." before Hindmarsh*

Done.

*P2 L23 Reformulate "temperature and surface mass balance perturbations diffusion"*

We removed the word "diffusion".

*P2 L26 New sentence with "GRISLI was in the late nineties ..."*

We have done so, thanks for the suggestion.

*P3 L4 Not only MISI, but also GL movement in general. Reformulate.*

We agree and simply replaced MISI by grounding line migration.

*P3 L15 "2.1 Ice thermo-mechanics". I didn't find the "thermo" aspect in this section. Reword?*

Section 2.1.6 deals with the temperature computation in GRISLI and the viscosity presented in Sec. 2.1.1 is temperature dependent. The expression of B_AT (and its temperature dependency) is now presented.

*P3 L21 Add equation number after "mass conservation equation".*

Added.

*P3 L23 "the vertically integrated velocities in x and y-direction u_x and u_y".*

Done.

*P4 L1 "where sigma ... and tau ... are the longitudinal and shearing stress tensor terms, respectively".*

It has been changed to:
"[…] where tau_ij=x,y,z are the shearing stress tensor terms and sigma_i=x,y,z the longitudinal stress tensor terms, defined as (i=x,y,z):
sigma_i = tau_ii"

*P4 15 Add an equation, explanation or reference how B_AT is calculated.*

We acknowledge that this information was missing in the original paper. We have added its equation.

This part has been re-written.

*P5 L5 "horizontal derivatives" and "vertical derivatives" of what? Clarify.*

Of velocity, information added.

*P6 L16 How is the water pressure defined? Add equation or reference.*

The hydrology model presentation has been largely expanded, including water pressure definition.

*P6 L25 So "beta = C*N" for temperate ice. What is assumed for the rest? Clarify.*

As mentioned in P5L26 of the original manuscript, for cold-based points we imposed a large basal drag that ensures no-slip condition. In addition, sliding velocities are set to 0 in this case. We have added this last precision in the manuscript:
"For cold-based grounded ice we impose a large enough basal drag (typically $10^5$ Pa) to ensure virtually no-slip conditions on the bedrock and the basal velocity is set to zero in this case."

*P6 L25 I expected to get some information on how C_f is calibrated in this paragraph, maybe just a list of options that could be used, then you get back to that later.*

At the end of the paragraph we have added:
"The Cf parameter will be part of the calibrated parameters in our large ensemble."

*P6 L29 Avoid use of "flux correction" as it has a specific meaning in coupling climate models. Use e.g. "flux calculation" as on top of the next page.*

Done.

*P7 L18-19 Here it would be good to already know how the grid is laid out. Where is the velocity defined compared to the ice thickness nodes. You later state that you are using a "staggered Arakawa C-grid", but it may not be clear to all what that implies. Could you add a clear description or figure where the different quantities are defined? This would also help in the following to explain the numerical implementation of certain schemes.*

We have added a figure presenting the horizontal staggered grids (Fig. 2), with a description of where the variables are calculated.

*P7 L19. So the flux is imposed on two grid points in both x and y direction and applied there simultaneously? More precision is needed to make clear how to implement this.*

We acknowledge that the description of the flux computation at the grounding line was somehow incomplete in its description. Relying on the new Fig. 2, we have explained better how the flux computed at the sub-grid position is affecting the actual velocity nodes. This now reads:
"In GRISLI, from the last grounded point in the direction of the flow, we compute the subgrid position of the grounding line in the x and y directions linearly interpolating the floatation criterion (dark green dots in Fig. 2). From this position, the flux at the grounding line is calculated using Eq. 17 or Eq. 18 (red arrows in Fig. 2). Because the flux at the sub-grid position is perpendicular to the local grounding line, ideally we should project this flux onto the x and y-axis. However, in the model, we assume that the grounding line is always perpendicular to either the x or y-axis (dashed brown line in Fig. 2). Similarly to Fürst (2013), the value of the flux qgl is linearly interpolated to the two closest downstream and upstream velocity grid points (dark blue arrows in Fig. 2) using the two bounding velocity points (light blue arrows in Fig. 2)."

*P7 L22 Wouldn't this give back force at the places where velocities are calculated, not at the GL where it is needed?*

Yes, the reviewer is perfectly right here. Unfortunately it is not obvious to infer the back force at the grounding line. For example, Pollard and DeConto (2012) and Pattyn (2017) evaluate the back force using the longitudinal stress at the first downstream point.

*P7 L27 Replace "Ice front" by "Iceberg" in front of "calving".*

Done.

*P8 L1 The section header states "Temperature coupling". There is no description of coupling in this section, only how the temperature is calculated. Maybe change title to "Thermodynamics" or "Ice temperature calculation".*

You are right. We have reformulated to "Ice temperature calculation" as suggested.

*P8 L2 Could give a sentence of introduction to state that this is the classic way to solve thermodynamics and similar to many other models (references). Or is there anything special here that I have overlooked?*

The referee is right here, this equation of advection-diffusion is common to most ice sheet models. The introductory sentence reads:

"Similarly to most large-scale ice sheet models (Winkelmann et al., 2011; Pollard and DeConto, 2012; Pattyn 2017), the temperature in GRISLI is computed by solving the general advection-diffusion equation of temperature: [...]"

*P8 L2- Consider to give some indication on how all of this is solved numerically. How are the differential equations discretised? Which numerical schemes are used for advection and diffusion (upwind, second order, Lax)?*

We have added this information in Sec. 2.3:
"For the temperature equation (Eq. 19), we solved a 1D advection-diffusion equation for each model grid point. The resolution is performed with an upwind semi-implicit scheme (vertical velocity and heat production at the previous time step is used). The ice thermal conductivity is computed as the geometric mean of the two neighbouring conductivity (Patankar, 1980). Because the horizontal diffusion is neglected, the only horizontal terms concern horizontal advection and are computed with an upwind explicit scheme. The heat production is computed at the velocity (staggered) grid points and is then summed up to the temperature (centred) grid points."

*P10 L12 Hardly any information is shared on how the given equations are solved numerically. I believe it would make the paper a much more interesting reference for other modellers and even people in your own group if some details would be added on the practical side of the modelling.*

In the revised version, we now explicitly show the staggered grid in Fig. 2. We have also added two additional figures in the supplementary material showing: i) the boundary conditions when extending the front to the edges of the geographical domain and ii) the matrix used to solve the elliptic equation. We have also considerably expanded Sec. 2.3, adding more information on the numerical resolution of the equations:

"The mass balance equation is solved as an advection-only equation with an upwind scheme in space and a semi-implicit scheme in time (velocities at the previous time step are used). The numerical resolution is performed with a point-relaxation method with a variable time step. The value of this time step is chosen to ensure that the matrix becomes strongly diagonal dominant to achieve convergence of the point-relaxation method. The criteria is thus a threshold that is inversely proportional to the fastest velocity on the whole grid. Note that this smaller time step is solely used for the mass conservation equation and subsequent variables (e.g. surface slopes, SIA velocity) while the rest of the model uses a main time step, typically ranging from 0.5 to 5 years depending on the horizontal resolution.

To solve it, the ice shelves/ice streams equation (Eq. 14) is linearised. The viscosity is computed using an iterative method starting from the viscosity calculated from strain rates from the previous time step. As this equation is the most expensive part of the model, the iteration mode is not always used depending on the type of experiment (for instance not crucial when the objective is to reach the steady state). In this case the viscosity of the previous time step is used. The linear system is solved with a direct method (Gaussian elimination, sgbsv in the Lapack library (www.netlib.org/lapack))."

*P10 L13 More precision is needed to understand what variables are defined where on which (staggered) grid, also in the vertical. How is the vertical grid laid out? Is the order up-down or down-up? Is the first vertical grid point from the top where T is solved assumed at the boundary or representing the middle of a first layer? How is that at the base?*

We have added a figure that presents a schematic representation of the horizontal staggered grids used in the model (Fig. 2). The direction of the vertical axis z is shown in Fig. 1 (pointing upward). In addition, we have added this information in the section on the numerical features:
"The model uses finite differences computed on a staggered Arakawa C-grid in the horizontal plane (Fig. 2). In the vertical, the model defines σ-reduced coordinates, $\varsigma=(S-z)/H$, for 21 evenly spaced vertical layers, with the z vertical axis pointing upward and $\varsigma$ downward (0 at the surface and 1 at the bottom). The coordinate triplet $(i,j,k)$ (in x, y and $\varsigma$ direction) is representative of the centre of the grid cell."
The first vertical layer ($\varsigma=0$) has its temperature in equilibrium with the annual mean surface air temperature (Dirichlet boundary condition). In turn, the last vertical layer ($\varsigma=1$) is either at the melting point (Dirichlet boundary condition) or receives heat from the bedrock below (Neumann boundary condition at the bottom of the grid cell).

*P10 L17 "the resolution is \*reduced\* to ...".*

Right, corrected.

*P10 L19 Replace "computes" by "uses" or "computes with". Add "which is dynamically calculated" after "(Eq.2)" or similar.*

We have followed your suggestions.

*P10 L24-29 Could this be visualised for clarity?*

A schematic representation of the matrix is shown in Fig. S1 in the supplementary material.

*P10 L28 How small is "small" in this context? Clarify*

We have a threshold on the integrated viscosity. Points tagged as "ghost" (see Sec. 3.2) have an integrated viscosity lower than 1500 Pa s. This paragraph has been reformulated.

*P11 L1 Add a reference for OpenMP.*

We provide an url redirecting to the OpenMP website.

*P11 L15 Add a short overview what is coming next before going into details.*

We have added at the very beginning of this section:
"In the following, we present a simple calibration methodology for the Antarctic ice sheet based on a large ensemble of model simulations."

*P11 L20 Add reference for ISMIP6 initMIP-Antarctica (ISMIP6 paper, website).*

Added reference to Nowicki et al., 2016.

*P12 L6 Is K0 changing the basal drag, or the basal drag coefficient? Clarify.*

The basal drag coefficient. We now refer more explicitly to variable names and equation numbers presented in Sec. 2.

*P12 L10 How does the BMB field keep the ice shelves stable? Clarify.*

For ISMIP6 initMIP-Antarctica, we use a data assimilation technique in which the basal drag coefficient is tuned to simulate an ice thickness as close as possible to observations assuming a fixed grounding line position. For ice shelves, we also inverse the basal melting rates under floating ice shelves so that the local (Eulerian) ice thickness derivative is minimal. The inferred basal melting rates are then averaged over the 18 ice shelf domains provided by the InitMIP project. With only a small addition, the sentence has been moved when first showing the present-day sub-shelf basal melting rates:

"Their values are based on the sectoral average of sub-shelf melt rates that ensured stable ice shelves (minimal Eulerian ice thickness derivative) in the recent intercomparison exercise InitMIP-Antarctica (Nowicki et al., 2016), with slight modifications due to change in resolution. They are in line with observations-based estimates (Rignot et al, 2013)."

*P12 L15 In my mind, the basic idea of LHS is to \*sample\* the hypercube and not perform all possible experiments. Maybe reword to avoid "the whole cube" if this is correct.*

It has been replaced to:
"We perform two times the 300 member ensemble with the flux at the grounding line of [...]".

*P12 L19 How is the low resolution data set produced from the original Bedmap2 data? Direct subsampling or smooth interpolation? This is a crucial part of preparing the input data and should be treated with detail and precision.*

We simply use a spatial bi-linear interpolation to generate the 40 km input data from the original high-resolution Bedmap2 data. This information has been added in the manuscript. We acknowledge the fact that this can alter the quality of the original data. In particular the shape (direction and slope) of fine scale structures such as narrow valleys and fjords might not be preserved. As stated in the discussion (second paragraph), we aim at introducing some sub-grid information in the model but this has yet to be done.

*P12 L20 Reword "discarded from the ensemble" to "not explored in this ensemble"*

Thanks for the suggestion, we have followed your suggestion.

*P13 L12 If differences are below 500, why does the scale go to 1000 in the figures?*

As stated in the manuscript, the differences are *generally* below 500 m although locally (e.g. Ronnie-Filchner area) we could have much larger differences.

*P13 L18 Do all these models use the same data, the same processing to get to the final input data and have similar resolution? Do you know if this is an error in the data or a problem of coarse resolution? What is different in models that do not show these features, if there are any?*

Pollard and DeConto (2009) also use a 40-km grid, while Martin et al. (2011) use their model at about 20 km resolution. However, input data, notably bedrock elevation / ice thickness and climate forcing, differ and can not explain why different models present systematic biases at the same locations.

The Ronnie-Filchner area presents a complex bedrock topography with pinning points stabilising the grounding line position. The fact that large scale ice sheet models do not explicitly account for sub-grid pinning points can explain consistent biases amongst the models. Similarly, the Transantarctic mountains present narrow ice streams draining part of the East Antarctic ice sheet and are generally poorly represented in large-scale ice sheet models.

More recent publications generally use an inverse method so that ice thickness mismatch with observations is greatly reduced and cannot be directly compared to the results presented in our manuscript.

*P13 L20 If the last point is resolved, maybe "... suggesting a common source of error related to the coarse model resolution". Or similar to add some interpretation to this comparison.*

We have added the following:
"Consistent model biases amongst these models, which use different input data, suggest a common source of error related to the coarse model resolution (20 to 40 km) or uncertainties in the bedrock dataset, particularly large in East Antarctica (Fretwell et al., 2013)."

*P16 L3 How can you be sure that the parameter range is sufficient/optimal for the transient experiments? Please add a short discussion on that.*

For the transient experiments, we used the twelve ensemble members that show the lowest RMSE when forced by perpetual present-day SMB. This shows the simulated variability through glacial-interglacial cycles for models calibrated on present-day ice sheet geometry. This is of course a too small subset to properly address the inter-model differences but we believe it is sufficient to discuss the general variability because the models produce similar evolutions. Ideally, we would have run all the ensemble members with the transient forcing and only kept the members showing the lowest RMSE at 0 kaBP. However, uncertainties in the climate forcing will also have biased the model responses.

We have nonetheless moderated the sentences:
"The uncertainty related to the choice of the internal parameters within our subset leads generally to up to [...]"

*P16 L10 Maybe "post-LGM retreat"?*

Thanks for the suggestion. Added.

*P17 L4 "relative to observations ... in this case, where ..." and briefly remind us what is different in the present experiments.*

This now reads:
"Whilst the model is able to reproduce present-day Greenland (Le clec'h et al., 2017) and Antarctic (Ritz et al., 2015) ice sheets when using an inverse method to estimate the basal drag, our simulations with an interactive basal drag computed from the effective pressure show some important disagreements relative to observations."

*P17 L5 Where is the northern part of East Antarctica? polewards = south!, north=towards the margin?*

We have reformulated as:

"In particular there are some persisting model biases in ice thickness. In East Antarctica, the ice thickness is underestimated towards the pole and the Transantarctic mountains while it is overestimated towards the margins, from Queen Maud land to Wilkes land. In West Antarctica, there is an underestimation of ice thickness in the Ronnie-Filchner basin and an overestimation in the Ross basin."

Changed to:

"However, by design, the fit with observations is systematically poorer compared to model results that make use of an inverse basal drag coefficient."

*P17 L23 This comes a bit unexpected. Has vertical temperature been worked on in this paper? Why only the vertical?*

For sake of clarity, we have been more specific in the introductory sentence:
"Although widely used for ice sheet model spin-up or calibration, long-term integrations under present-day forcing induce a warm bias in the vertical temperature profile because they discard the diffusion of glacial-interglacial changes in surface temperature."

In our paper, the calibration step has been done assuming a perpetual present-day climate forcing, which necessarily bias our simulated temperature field towards too high value. We believe that this point deserves discussion in this section and we have kept it in the revised manuscript.

*P17 L33 "lead to a more dynamic grounding line position"*

We have followed your suggestion.

*P17 L34 Has sensitivity to SL changes really been discussed in the paper?*

We have not tested explicitly the sensitivity to sea level change in this paper but it has been accounted for when performing glacial-interglacial simulations.

We now mention the sub-shelf melt rates. However, our point here was about changes in sea level and its local variations as this has been relatively unexplored within large-scale ice sheet model while sub-shelf melt rates is known to be a driver for glacial-interglacial Antarctic variability.

*P18 L11 Basal hydrology and semi-lagrangian tracking are described in "2.2 Additional features" next to other aspects that are not mentioned here in the conclusions. It may be useful to make it clearer already in the main text which of the described features are new developments in GRISLI.*

The "additional features" section contains material that is not directly related to the ice model itself (ice sheet thermo-mechanics). In this section, there is indeed old and new model features.

The basal hydrology has only been documented in a Phd dissertation written in French (Peyaud, 2006). The semi-lagrangian tracking has been presented by Lhomme et al. (2005) but for an other

ice sheet model. It has been re-implemented in GRISLI for Quiquet et al. (2013) but has not been described there.

In the revised version of the manuscript we have listed better the difference with Ritz et al. (2001). Following is a list of the additions in the revised manuscript:

Introduction
"We also provide details on some components (sub-glacial hydrology and tracking particle scheme embedded in GRISLI) which are currently not documented in international scientific journals."

Section 2.1.1
"Ice deformation and mass conservation in GRISLI version 2.0 is mostly treated as in Ritz et al. (2001) with the notable exception of the use of a polynomial flow law with the introduction of a linear, Newtonian, viscosity."

Section 2.1.2
"Differing from Ritz et al. (2001), the velocity in GRISLI is now computed for the entire domain as the superposition of the shallow ice approximation (SIA) and the shallow shelf approximation (SSA) components, without using a sliding law to estimate basal velocities."

Section 2.1.6
"The ice temperature calculation has remained identical to Ritz et al. (2001)."

Section 2.2.1
"In the following we provide a complete description of the hydrology model because it has only been described in a French Phd dissertation (Peyaud, 2006) but currently lacks a description in an international scientific journal."

Section 2.2.2
"As in Ritz et al. (2001), GRISLI computes the bedrock response to ice load with an [...]".

Section 2.2.3
"GRISLI includes a passive tracer model that allows for the computation of vertical ice stratigraphy, i.e. time and location of ice deposition for the vertical model grid points. The model is the one of Lhomme et al. (2005) re-implemented in GRISLI by Quiquet et al. (2013)."

*P18 L26 Why does validating a model for the Antarctic give confidence to also use it for the NH. A bit more information is needed here to bridge that gap.*

If we consider today ice sheets that are still present now, the Antarctic ice sheet is a good case study because of its interaction with the ocean which could have been more important through glacial-interglacial cycles compared to the Greenland ice sheet. The fact that we manage to reproduce the Antarctic ice sheet variability gives some confidence in the ability of the model to simulate large grounding line migration that could have happened for palaeo ice sheets such as for example the Kara-Barents ice sheet. However, sub-glacial conditions (e.g. till distribution and rheology) are probably largely different and this has not been explored in the present manuscript. To avoid overstated statement we have decided to remove this sentence from the revised manuscript.

*P18 L30 Replace "and" by "or", unless all three have to be contacted to get the model code.*

Changed to "or", no need to contact all three.

*P26 in the middle. Replace "N.m" by "N m"?*

Yes, changed.

*P27 caption Table 2. Write out LHS.*

Done.

*P27 Figure 2. Left panel cold have additional contours to delineate the regions and a colour bar. Why does the grounding line have different melting rates than the shelf?*

The left panel simply displays the different sectors as defined in ISMIP6 InitMIP-Antarctica (Nowicki et al., 2016). The numbering is irrelevant as we never refer to the sector number in the text. We have nonetheless changed the color coding so as to show better the extent of the different sectors.

We have added this in the figure caption:
"The melting rates are different for the shelf and the associated grounding line to mimic the higher values observed close to the grounding line Rignot et al. (2013)."

*P29 Figure 5. Use a colour for the GL contour that does not appear in the colour map, e.g. black or dark gray.*

Thanks for the suggestion, we have used dark grey instead of red.

*P30 Figure 6. Same as for figure 5.*

Changed here as well.

*P32 Figure 8. Suggest to move the parameter values to a table or the main text.*

We have moved this information to a table.

*P34 Figure 12. "materialised" –> "shown" or "indicated" Why is the GL so patchy here compared to the steady state case?*

We have chosen the word "indicated".

Fig. 5 and Fig. 6 in the original manuscript shows only the present-day observed grounding line. But in fact the grounding line is also relatively patchy for the steady state as soon as the model simulates a significant departure from present-day position. You will find below the surface elevation for the two ensemble members that show the lowest RMSE in Sec. 3 (100-kyr simulation under perpetual climate) in which the grounding line is indicated by a thick red line.

[Figure]

**Figure R1:** Simulated surface elevation at the end of the 100 kyrs simulations forced by perpetual present-day climate (Sec. 3) for the two ensemble members that show the lowest RMSE with respect to the observations (AN40S123 and AN40T213). The grounding line is indicated by the thick red line.

*P35 Figure 13. Same as for figure 5.*

Changed here as well.

---

## Author Response (AR2)

**Comments to the Author:**

*Thank you for your revised manuscript. I am sorry for the delay, which was caused by the previous topical editor no longer being available.*

*I think that you have answered most of the reviewers concerns, but there are still some problems with the manuscript.*

Thank you for your time and effort to improve our manuscript. Please find below our response and a revised version of the manuscript.

*Language...*

*The response to the reviewers and the revised manuscript are both very difficult to understand. I think the main problem is incoherence of language rather than incoherence of thought, but I cannot accept this manuscript until I have seen a version with much improved English.*

*Some of the problems:*
*Words are spelled incorrectly;*
*Words are used inappropriately;*
*Tenses are wrong;*
*Singular and plural are mixed up;*
*Punctuation is missing;*
*It is often not clear what "it" refers to;*
*Other grammatical errors that render sentences incomprehensible;*
*Several typos.*

*I will send you my copy of your response in which I have highlighted all the parts that I had to read multiple times in order to understand your meaning. Doing this helped me to understand your work. I hope it will also help you to improve the manuscript.*

*Note that "similar" means "resembling without being identical". If you write that something is similar to something else then you also need to mention in which ways it is different. If you actually mean identical/the same, then say so.*

*For example,*
*"Amongst these parameters, along with the basal effective pressure, the large scale bedrock curvature and/or sub-grid roughness could be used, similarly to Briggs et al. (2013)." could become,*
*"Amongst these parameters, along with the basal effective pressure, the large scale bedrock curvature and/or sub-grid roughness could be used, as in Briggs et al. (2013).", should this be the meaning that you intend.*

*Note, also, RMSE is not yielded or presented.*
*"... have an RMSE of ...", is sufficient.*

None of the authors are English native and even with multiple thorough readings it makes it difficult for us to provide an error-free manuscript. We thank you for your time highlighting our mistakes, we did our best to provide a corrected revision in this new version.

*On to the science…*

*Section 3 "Calibration of the Antarctic ice sheet"*

*This section is unconvincing. I think an additional paragraph is needed here, to explain what you are doing and why. What I would expect to see for a model calibration is the development of a cost*

*function that specifies a PDF of what you believe to be plausible models. The final set of models is then sampled from this PDF. In order to develop this PDF, discussion of the observations, the observational error, and the model adequacy (or discrepancy) need to be included.*

*What you have actually done seems rather bizarre. You appear to have 3 observations (total volume, ice thickness, ice velocity), and yet you have used only one of these (thickness). You have included in your final ensemble only those models with the lowest RMSE compared to that single metric. Observational error is not mentioned. Discussion of model (or in this case experimental) adequacy suddenly appears in a paragraph of the Discussion (the one that starts "Although widely used... ") where you explain that you do not anyway believe the result of your calibration. I conclude from this that you in no way believe that your 12 "best" ensemble members are in fact your "best"! Would you be planning to use this set of 12 in any future work?*

*I might be being stupid, but I cannot find figures showing the observed thickness, or uncertainty in that thickness. In addition the plots of RMSE are not very informative as they do not indicate whether the ice is too thin or too thick. I also cannot tell whether (and where) the whole ensemble is biased, which is surely important information.*

*I wonder whether what you are doing is more of a sensitivity test than a calibration. It seems to me that what you are investigating is the range of model behaviours after a strong constraint is placed on ice thickness. If so, then the section needs quite a bit of rewriting to make this clear.*

**On the choice of the metric:**

We have added a new paragraph to the new version of the manuscript to justify our choices.

it is true that we discuss three possible ways to constrain the model: ice volume, ice thickness and ice velocity. However, these three are not equivalent for the purpose of calibrating ice sheet models.

Ice velocities are a widely used target for ice sheet models, in particular when using inverse methods to tune ice dynamics. In this case, ice geometry (ice thickness) is generally prescribed to its present-day value and a dynamical parameter (e.g. basal drag coefficient) is tuned to reproduce the ice velocity field (e.g. Price et al., 2010; Gillet-Chaulet et al., 2012).

When no inversion methodology is used, a useful metric based on ice velocities is more complicated to define:
- Ice velocities are spatially highly variable and present their maximum values at the ice sheet margins. This means that small errors in the simulated extent of the ice sheet lead to important discrepancy with the observations. As such, marginal regions, which represent a small fraction of the ice sheet, have more weigh for metrics such as the RMSE. For example: having the shape of a simulated ice shelf slightly different from the observations is not crucial for ice dynamics but have important consequences on a RMSE using ice velocities.
- In addition, ice velocities show fine scale structure resulting from the complex bedrock below the ice sheet (together with geologic bed properties, hydrological and thermal conditions). Our coarse-resolution model (40 km) is not suited to reproduce the fine scale structures and can only reproduce the broad pattern.
- Finally, aggregation of fine scale (~100 m) ice streams with neighbouring slow ice is not meaningful at 40 km resolution.
For all these reasons, ice velocities are in general not used as a tuning target for large spatial scale ice sheet simulations at coarse resolution.

Conversely, ice sheet geometry (in our case, assessed by ice thickness RMSE) is usually a primary target for ice sheet modellers (e.g. Ritz et al., 1997; Stone et al., 2010; Applegate et al., 2012; Pollard et al., 2016). The advantage of ice geometry is that it is relatively well constrained for present-day but also in the past (ice margin migrations from geological evidence, e.g. DATED and

RAISED community effort). This means that ice sheet models can sometimes be validated with respect to our knowledge on palaeo-geometries.

Ice volume is the spatial integration of ice thickness and is generally a poor metric to assess model performance due to potential compensatory biases. In turns, ice thickness RMSE is a more robust metric.

For all these reasons, we follow the approach commonly adopted by the community building our main metric on ice geometry. The justification of the approach at the end of Sec. 3.1 reads:

"In the following, individual member performance is assessed with the root mean square error (RMSE) computed from simulated and observed ice thickness (Fretwell et al., 2013). This metric puts a strong constraints on ice sheet geometry and avoids potential compensatory biases that could appear when using total ice volume. Observed ice velocities are not used as a metric because of high spatial variability that results from small-scale bed properties. With a coarse 40-km resolution, the model is intrinsically unable to reproduce such variability but is expected to reproduce the large scale pattern of ice thickness. However, at the end of the long 100-kyrs simulation, the velocities in the model are the balance velocities corresponding to the simulated topography. Thus, the minimisation of the RMSE in ice thickness should also reduce the error in velocities with respect to observations at the global scale."

**On observational error:**

Following your advice we added a figure showing the ice thickness observational dataset together with its uncertainty (Fretwell et al., 2013). Fretwell et al. estimate generally an uncertainty of about $\pm150$ m. However, there are some regions for which the lack of in-situ measurements forces the authors to use gravity-derived ice thickness estimates only. The estimated uncertainty then reaches ±1000 m. We agree that the addition of this figure helps the reader to interpret our model results. We explicitly refer to this figure when commenting the model bias (new reference in Sec. 3.2).

**On model biases:**

The response of the model within our large ensemble presents both positive and negative volume anomalies, with members overestimating (about +38% within AN40S and +27% within AN40T) and members underestimating (about -27%) total ice volume. This is shown in Fig. 5 and Fig. 6. Our ensemble is thus not biased towards smaller/larger ice sheets. However, as explained in the manuscript (Sec. 3.2), we have persisting model biases in some regions (East Antarctica and Ronnie-Filchner basin), as shown in Fig. 7 and Fig. 8.

**On the usefulness of the parameters yielded during the calibration:**

In the discussion we mention that a calibration based on glacial-interglacial simulations is ideally preferred to the approach followed in Sec. 3.1 (perpetual modern climate). However, the determination of a realistic climate forcing for palaeo-simulations is a considerable challenge and mechanical parameters inferred with such an approach will necessarily reflect the assumptions made on past climate change. Even though the parameter values found in Sec. 3.1  minimising the RMSE do not take into account past climate change, they are nonetheless useful since, as shown in Sec. 3.2, they are able to provide glacial-interglacial ice geometry changes that roughly fit available palaeo-data constraints. To emphasize this point, we added the following in the discussion:
"[…] behaviour for long-term integrations. Using the parameters calibrated under perpetual modern climate, the model is nonetheless able to reproduce ice geometry changes compatible with palaeo-constraints. Further work [...]"

*Numerical solution methods…*

*I have rarely seen a manuscript in which the reviewers clamour so loudly for the code to be made available. While some description of the numerical solution methods has been added to the manuscript, it is still rather generic (eg "an upwind scheme" !!) and I think this is insufficient for GMD's aim of "scientific reproducibility". One possibility would be writing the difference schemes in an appendix or in the supplement, but this is unlikely to fully satisfy all readers, and a simpler solution would be to be more specific in the manuscript while also making the code developed in this paper available in the supplement.*

*I do not fully understand the grounds under which the code is (or is not) being made available at present. You wrote in your response,*
*"The public distribution needs to be done with a specific license (e.g. GNU public licence). Such a license is currently lacking in GRISLI. The model has been developed from the 90ies and has benefited from the additions from numerous contributors. To put a license to the current model, we have to get the permissions to all the contributors from the past 20 years. To date, we do not have this permission from all the contributors. However, as stated in the manuscript, any potential users are encouraged to get in touch with C. Dumas, A. Quiquet or C. Ritz, to start a collaboration."*

*My understanding is that it is the employers, rather than the individuals, who need to give permission. How many of these employers have you been unable to track down (are some no longer active?), and which parts of the code are still restricted? If you upload all parts of the code that are not restricted, will this include the parts of the code that the reviewers want to see (ie the numerical solution to the equations outlined in this manuscript). If so then this may be satisfactory solution for the present manuscript. It seems strange that you can grant access upon request, but not release the code - what is the current license that users get when given access to the repository? I do understand that it can take a long time (years!) to achieve full code access for older models, but I would like to see some evidence in the manuscript that this is being actively pursued. It is already foreseeable that in the future all code developed in scientific publications will have to be made publicly available, so you are strongly encouraged to make this a reality for your own model sooner rather than later.*

Encouraged by referee #3 comment and yours, we started to put parts of the code under the CeCILL public licence v 2.1 which is our employer's requirement for compatibility with GNU/GPL. Previously requested code sections are attached to the present revision of the model and are intended to be public. Sections of the code currently under the CeCILL licence include:
- Resolution of the elliptic equation
- Implementation of Schoof (2007) and Tsai et al. (2015) in the model
- Basal hydrology
- Passive tracer
We hope that these files cover most of the critical sections of the model and will be helpful for the readers.
We are currently working to release the whole model code under the CeCILL license. This however is work in progress and will require additional work. We will update you with the status of our efforts by the end of the review process in order to make a final version of the code availability section.

[revised manuscript text omitted]

---

## Author Response (AR3)

**Comments to the Author:**

*Thank you for the revised manuscript. It is amazing how much a manuscript improves when even only a few of the worst language mistakes are corrected (and you have done much more than such a minimum!). There is an obvious typo in the abstract, but I understand the manuscript well enough now to leave the rest to the copy editors (their English is much better than mine anyway).*

*The outstanding problem is the calibration. I asked for an introductory paragraph explaining the purpose of this, and I still think it is required because I do not know what the point of the exercise is. If the aim is to calibrate the model why run on 12 members? If the aim is to look at uncertainty, why run the 12 that are closest to the data? The data look like they have fairly high uncertainty - what range of RMSE might you expect from a perfect model? Are the 12 ensemble members all within or outside of that range? Usually, when looking at uncertainty, the aim is to explore some range of plausible models. There is always quite a bit of subjectivity involved, but something people might do is loosely re-weight the ensemble by the data (including effects of model and data error) and then resample from that re-weighted ensemble. The reason it is important to explain this much more clearly is so that your results (and the ensemble members themselves) may be used appropriately in future. Also, without this guidance the reader does not know what conclusion to draw from Figure 14.*

The calibration is presented in Sec. 3. For this, we run 600 ensemble members (300 for each analytical formulation of the flux at the grounding line) for 100 kyrs. Ensemble member performance is assessed with one metric (RMSE on ice thickness). The ensemble members that show the lowest RMSE are considered as the best models (best representation of present-day ice sheet topography) and can be used for other applications.

As explained in the third paragraph of the discussion, we do not use transient simulations of the glacial-interglacial for calibration because the definition of climatic boundary conditions for the ice sheet model is a considerable challenge given the many degrees of freedom. That is why the calibration is done in a more constrained framework, i.e. present-day conditions. However, by construction, equilibrium simulations such as the ones in Sec. 3 do not allow for the validation of the dynamical response of the flux at the grounding line since there are no climatic transitions and subsequent migration of the grounding line. The aim of Sec. 4 is to verify that the new formulations of the flux at the grounding line allows for grounding line migrations compatible with palaeo-data constraints. Again, the aim of Sec. 4 is not to calibrate the model since the calibration is performed in Sec. 3.

In Sec. 4 we construct a simple climatic forcing and use 24 plausible models (12 ensemble members with lowest RMSE within AN40S and AN40T) to simulate the Antarctic ice sheet for the last 400 kyrs. Even if the range of RMSE is small within this subset, they have very different parameter values (Fig. 5 and Fig. 6). We could have run the transient simulation with only one model (e.g. the one with the lowest RMSE). However, using these 24 plausible models can help the reader to grasp the GRISLI result spread (Fig. 14) for models yielding a relatively similar present-day ice sheet. We acknowledge that the choice of 12x2 ensemble members is arbitrary and this number is too low to infer statistically meaningful results in terms of model uncertainty. However, even with our relatively coarse resolution, a 400 kyr-simulation represent a non-negligible computing time that has to be added to the 600 ensemble members of Sec. 3.

As a result, Fig. 14 aims at illustrating the spread in ice volume within the 12x2 ensemble members calibrated to reproduce the present-day Antarctic ice thickness.

Motivated by your comment, we added a concluding paragraph at the end of the calibration section (Sec. 3.2) to explain better how the 12 ensemble members are used:

"*From the 300 ensemble members within AN40 and AN40T, we keep the 12 ensemble members that have the lowest RMSE. We consider that this small subset contains plausible models for the Antarctic ice sheet. These models are used in the next section for transient simulations covering*

*the last 400 kyrs. While they have a similar RMSE, they have distinct parameter values (Fig. 5 and Fig. 6) and as such they provide an insight of the uncertainties in ice sheet evolution relative to parameter choice.*"

We also better explain the aim of the transient experiments (introductory paragraph of Sec. 4):

"*By construction, equilibrium simulations such as the ones shown in Sec. 3 do not allow for the validation of the dynamical response of the flux at the grounding line since there are no climatic transitions and subsequent migration of the grounding line.* The main objective of this section is *thus* to show the ability of the model to reproduce large ice sheet geometry changes in response to Quaternary climate change."

We also added in the abstract:

"*To assess the ability of the model to simulate grounding line migration,* we also present glacial-interglacial ice sheet changes throughout the last 400 kyr using the best ensemble members taking advantage of the capacity of the model to perform multi-millenial long-term integrations."

We hope that these additions will help the reader to understand our methodology.

Finally, note that the code availability section has been amended to reflect that part of the GRISLI code is available in the supplement. We refer to this in the text as well:

Sec. 2.1.2:

[revised manuscript text omitted]

---

## Author Response (AR4)

Thank you once again for your comments. We are not 100% confident that we have understood your exact point. We have tried to answer your concerns in the following.

To your first concern:

*There has not however been much success in answering my questions about the calibration apart from calling the ensemble member "plausible", without giving any reasons why you think this is the case. I think you might have misunderstood what I am asking. I am not asking why you do not use transient experiments for your calibration, neither am I asking for more details on what was actually done. Rather, it is necessary for you to explain the basis for the calibration experiment and what the resulting ensemble means and therefore what it is useful for. I still think that an additional paragraph to this end at the start of section 3 is required - it is confusing to jump straight from the calibration subtitle into the method subtitle without explaining why you are doing it. I am sure this stuff is inside the heads of you and your co-authors, but readers are not telepathic - I just want it written down so that readers can understand the purpose of your work!*

We have modified the text of the manuscript at the start of section 3 as follow:

**3 Calibration for the Antarctic ice sheet**

Over the years, several GRISLI internal parameters have been shown to be of importance to appropriately simulate the flow and mass balance of the Antarctic ice-sheet. Values for these parameters have been so far derived from non-systematic tests and expert knowledge. To systematically investigate the role of those parameters and find the best fitting set for the simulated Antarctic ice-sheet with respect to the observed one, a calibration methodology with systematic exploration of the different values is performed in the following. The best fitting set will be considered as plausible models within the chosen parameter space.

 Given its degree of complexity, GRISLI is mostly designed for multi-millenial integrations. Due to long-term diffusive response to SMB and temperature changes, an accurate methodology to select unknown parameters of the model would be to run long transient simulations with a climate forcing as close as possible from past climate states, ideally with a synchronous coupling between the ice sheet and the atmosphere. However, climate models generally fail at reproducing the regional climate changes during the last glacial-interglacial cycle as recorded by proxy data (Braconnot et al., 2012). Further- more, the phase III of the Paleoclimate Modelling Intercomparison Project (PMIP3) has highlighted the large disagreement between participating climate models in simulating the Last Glacial Maximum (LGM) in the vicinity of northern Hemisphere ice sheets (e.g. Harrison et al., 2014). Given these uncertainties amongst climate models and the large sensitivity of the ice sheet model to climate forcing fields (e.g. Charbit et al., 2007; Quiquet et al., 2012; Yan et al., 2013), it is difficult to calibrate the mechanical parameters independently from that of the SMB, in particular for northern Hemisphere ice sheets.

For these reasons, here we suggest a simple calibration methodology for the Antarctic ice sheet in which the model is run for 100 kyrs under a perpetual modern climate forcing until equilibrium.

**3.1 Methods**

 In the following, we use the 27 km-grid atmospheric outputs, namely annual mean temperature and SMB, from the regional climate model RACMO2.3 (Van Wessem et al., 2014), averaged over the 1976-2016 time span. The basal melting rates under ice shelves are prescribed for the 18 sectors of the Antarctic ice sheet as defined in ISMIP-Antarctica project (Nowicki et al., 2016) and are shown in Fig. 3. Their values are based on the sectoral average of sub-shelf melt rates that ensured stable ice shelves (minimal Eulerian ice thickness derivative) in the recent intercomparison exercise InitMIP-Antarctica (Nowicki et al., 2016), with slight modifi- cations due to change in resolution. They are in line with observations-based estimates (Rignot et al., 2013). We do not apply any correction related to geometry changes to the climatic forcings during the calibration.

To your second concern:

*I previously suggested that perhaps your calibration+transient ensemble experiment was closer to being a sensitivity test, as it shows that model versions with similar fit to the data can have different parameters and quite different results. So now I am now very confused, because your latest Author's response appears to support this line of reasoning, but the text of the manuscript does not! Maybe I have missed a paragraph in the manuscript (in which case I am sorry and please point it out to me) but I can't see where you include these points from your own response, "However, using these 24 plausible models can help the reader to grasp the GRISLI result spread (Fig. 14) for models yielding a relatively similar present-day ice sheet. We acknowledge that the choice of 12x2 ensemble members is arbitrary and this number is too low to infer statistically meaningful results in terms of model uncertainty. However, even with our relatively coarse resolution, a 400 kyr-simulation represent a nonnegligible computing time that has to be added to the 600 ensemble members of Sec. 3. As a result, Fig. 14 aims at illustrating the spread in ice volume within the 12x2 ensemble members calibrated to reproduce the present-day Antarctic ice thickness."*

We have added this notion before the start of section 4 as follow:

From each of the two ensemble (AN40S and AN40T), we keep the 12 ensemble members out of 300 that have the lowest RMSE and use them in the next section for transient simulations covering the last 400 kyrs. Using these 24 plausible models on long term transient integration provides insight on the GRISLI result spread for models yielding a similar present-day ice sheet. Indeed, while they have a similar RMSE, they have distinct parameter values (Fig. 5 and Fig. 6) and as such they provide an insight of the uncertainties in ice sheet evolution relative to parameter choice. We acknowledge that the choice of 12x2 ensemble members is arbitrary and this number is too low to infer statistically meaningful results in terms of model uncertainty. Still, even with our relatively coarse resolution, 400 kyr-simulations represent a non-negligible computing time that has to be added to the 600 ensemble members of Sec. 3.

**4 Antarctic ice sheet changes for the last 400 kyrs**

**4.1 Methods**

By construction, equilibrium simulations such as the ones shown in Sec. 3 do not allow for the validation of the dynamical response of the flux at the grounding line since there are no climatic transitions and subsequent migration of the grounding line. The main objective of this section is thus to show the ability of the model to reproduce large ice sheet geometry changes in response to Quaternary climate change. As a consequence of our limited knowledge of past climatic conditions in the Antarctic ice sheet region over glacial-interglacial cycles, we use here an idealised reconstruction of SMB, near surface air temperature and oceanic basal melting rates based on a limited number of proxy records. Our approach is somewhat similar to previous works (e.g. Ritz et al., 2001; Huybrechts, 2002; Pollard and DeConto, 2009; Greve et al., 2011; Golledge et al., 2014).